# A Multi-Modal Profiling Fraud-Detection System for Capturing Suspicious Airline Ticket Activities

**Mehmed Taha Aras** [1,†] and **Mehmet Amac Guvensan** [1,*,†]

Department of Computer Engineering, Yildiz Technical University, 34349 Istanbul, Türkiye; m.taha.aras@gmail.com

* Correspondence: amac@yildiz.edu.tr
† These authors contributed equally to this work.

**Abstract:** Although the most widely studied datasets in fraud-detection systems belong to the banking sector, the aviation industry is susceptible to fraud activities that seriously harm airline companies. Therefore, big airline companies have started to purchase or develop their own fraud-detection systems in order to prevent their financial loss and prestige decline. Chronological order and temporal flow are intrinsically of high importance in fraud detection in the banking sector as well as in airline sale channels. Therefore, the transactions in the datasets used in fraud-detection systems should be evaluated not only according to the information they contain but also according to the past transactions they are linked to. One of the best ways to raise awareness about the connected past transactions to the fraud-detection system is to profile the data fields whose historical data is important and dynamically place these profiles on each transaction. In this study, we first draw the baseline, i.e., the first touch in this field, for fraud detection in aviation and then introduce a novel multi-modal profiling mechanism based on deep learning for the detection of fraudulent airline ticket activities. We achieved great success by feeding the new features obtained from those profiles into a deep neural network that is fine-tuned by adjusting the well-known hyperparameters regarding the aviation data. Thanks to the combination of profiling and deep learning, the F1 score of the proposed system reaches up to 89.3% and 93.2% in terms of quantity-based success and cost-based success, respectively.

**Keywords:** fraud detection; aviation industry; profile-based systems; machine learning; deep learning; imbalanced dataset; cost-sensitive measurement; history-aware detection



## 1. Introduction

In recent periods, the amount of online shopping and payment has been increasing at a great pace. Following this trend, online sales have become the new target of fraudsters in the last decade. Thus, ongoing evolutionary fraud attempts always cause companies to experience serious losses both financially and in terms of customer satisfaction. The aviation industry is one of the well-known sectors for online ticket sales where the most losses are experienced in terms of both financial and reputation due to fraudulent activities. In accordance with civil aviation laws, the airline company has to compensate the loss of the person whose credit card is stolen and used in the purchase of airline tickets or in-flight services. Therefore, airline companies lose millions of dollars each year and/or miss potential customers. To minimize their losses, such companies develop or purchase fraud-detection systems to secure their sale channels and prevent fraudulent attempts. Analyzing the market shows us that the majority of these fraud-detection systems generally exploit traditional rule-based mechanisms [1,2]. Unfortunately, rule-based fraud-detection systems are incapable of adapting to the rapidly changing strategies of fraudsters and revealing complex fraud patterns. Therefore, instead of traditional rule-based systems, automated systems that can detect fraud using machine learning or deep learning methods

have become prominent [3,4]. However, these learning methods alone do not provide the desired effectiveness for an ideal fraud-detection system. Fraud detection is a complex issue that needs to be focused on many different points [5] and problems, especially regarding the sector facts and dynamics.

One of the main problems for fraud-detection research in the literature is that the well-known datasets consist of individual transactions. However, since fraud is a temporal and chronological problem, the historical links of the transaction are also of great importance. Information about a single transaction may not always be sufficient to detect fraud. Hence, when evaluating a transaction, possessing information about its associated previous transactions enhances the historical robustness of the fraud-detection system. While there can be multiple avenues to leverage this strength, one of the most effective approaches involves transforming traditional systems into profile-based solutions. This transformation is achieved by crafting substantial profiles that can directly influence the detection mechanism through the utilization of historical data associated with each profile. In this study, we introduce a novel profile-based fraud-detection system for suspicious airline ticket activities in order to differentiate a fraudster from an ordinary airline passenger using his/her past activities. In this system, we put forward various historical and in-record profiles by introducing discriminative time-line features that have a direct impact on the success of fraud detection in the airline industry. These additional fields make it possible for each transaction to be aware of the related transactions before itself. In addition, the combinations of these designed profiles were analyzed in order to reveal the superior profile combination for the aviation field.

From the perspective of companies exposed to fraud cases, fraud is a financial problem. Therefore, the primary purpose of developing fraud-detection systems is to reduce financial losses and prevent potential decreases in revenue. However, in most studies [6,7], the success results are calculated based on the quantity for both traditional and automated detection systems. These measurement metrics, which are based on the number of false/true detected transactions, cannot show the performance of fraud-detection systems thoroughly. For this reason, in our study, besides the transaction number-based metrics, the cost-based metrics were utilized for the evaluation of the proposed profile-based architecture. In this way, it is also possible to observe to what extent the developed detection system can achieve loss minimization and potential profit optimization. The ability of the detection system to produce both quantity-based and cost-based results is an important criterion in terms of maintaining the balance between company profitability and customer satisfaction.

The scientific goals of the fraud-detection system developed in our study can be summarized in two essential points. First of all, we aim at maximizing the F1 score, which is the most important performance metric for such imbalanced problems. Since a single and static profile structure ends up with low success in fraud detection and with many false positives, we build up historical connections using existing passenger activities obtained from the proposed profiles via feature engineering. The second goal of our study is to overcome the negative effect of the randomly selected instances, as they might not represent the non-fraud data in an efficient manner. Thus, we introduce the BRUS algorithm to include the most representative instances from the airline ticket dataset. This approach has a high negative sample representation ability as well as oversampling, has no misleading synthetic positive record, and provides ease of processing thanks to its smaller size.

Our study makes the following contributions to the literature:

- To the best of our knowledge, there is no study in the literature for the aviation industry that provides solutions to fraud-detection problems and builds detection systems using modern supervised learning techniques. In this study, for the first time in the aviation field, a baseline was created using one of the biggest airline activity datasets with a specific data structure and supervised machine learning (ML) algorithms. After we created the baseline and examined the ML results, we applied deep neural network

(DNN) as our main classification method to enhance the overall performance of the system. We also customized the DNN architecture utilizing different hyperparameters based on aviation data to boost its performance.

- Analyzing passenger activities encouraged us to develop a multi-modal profile-based fraud-detection system in order to get rid of the restrictive structure of transactional datasets in terms of backward connection. This aviation-specific fraud-detection system can establish historical connections by taking power from three different profiles and many statistical parameters within these profiles. Therefore, it significantly increases the success of fraud detection, which is a temporal problem.

- Fraud is essentially a financial problem in terms of its consequences. However, considering this problem in the literature, the number of studies using cost-sensitive measurement metrics is quite low. Due to civil aviation laws, airline companies have to regulate customer grievances, so the dimension of financial loss comes to the fore. The success of the fraud-detection system developed in this study was measured not only on the basis of quantity, but also with the cost-sensitive metrics that we recommend, and we tried to achieve a balanced success.

- The balanced random undersampling (BRUS) method has been developed to solve the imbalanced dataset problem, which is the most frequently encountered problem in the detection of fraud. In this method, there is a mechanism that evenly distributes the dataset and prevents duplication during sample selection. This method was also utilized as the first step in hybrid sampling tests in our study.

Section 2 presents the literature review on fraud detection. Section 3 explains our dataset and aviation terminology in general terms. In Section 4, common challenges in fraud detection and the key points of these challenges are specified. In Section 5, we introduce the proposed profile-based fraud-detection mechanism and describe it in detail. Section 6 gives all the technical steps of the proposed fraud-detection system. The results of all experiments are demonstrated and discussed in Section 7. Final comments and inferences both about the developed detection system and the proposed approaches are given in Section 8.

## 2. Literature Review

Fraud cases have been emerging in many sectors, especially with the broad usage of online payment. Most of the studies have been carried out in the banking sector, in which different approaches are recommended to develop successful fraud-detection systems and solve certain fraud-detection problems, such as [8–12]. Since the common gateways of sales transactions are made through virtual POS machines belonging to the banks, the frequency of fraud may be higher than in other sectors. The e-commerce sector, where online payment transactions take place intensively, is also among the areas where fraud-detection studies are carried out [13–15]. Since large payments can be made in the insurance sector, fraud cases can be seen frequently. Thus, various fraud-prevention studies such as [16–18] have been carried out to proactively prevent insurance abuse. Although the aviation sector is one of the sectors most affected by fraud cases, it is not covered much in the literature. Cybersource is a technology provider from which airline companies purchase products or services for fraud detection [19]. However, since these currently used fraud-detection products are rule-based, they have low effectiveness and sustainability. Fraud-detection studies in the literature can be examined in three main scopes, including traditional rule-based systems, modern automated systems, and systems using history-aware detection techniques.

Some important problems in most fraud-detection systems need to be solved. The most important one is the imbalanced dataset problem. Sampling techniques are frequently used approaches to solve the imbalanced dataset problem [20,21]. In some studies, hybrid methods are suggested by modifying sampling techniques in different ways. Hanskunatai [22] created a three-phase hybrid sampling mechanism using DBSCAN for clustering and undersampling/oversampling techniques for sampling. When this hybrid mechanism

is used, an increase of up to 69.91% according to the SMOTE algorithm, up to 19.34% according to the Tomek Links algorithm, and up to 59.93% according to the non-sampling situation was observed on the basis of F1 scores in tests performed on different datasets. The concept of temporality inherent in fraud detection is not addressed in this study, so transactions in some time periods of the dataset may be lost during undersampling. In addition, no direct or indirect precautions were taken for the problem of non-selection of linked transactions that occurs during the undersampling stage. Finally, using only the accuracy metric as a success criterion in the field of fraud, where the imbalanced dataset problem occurs, is a misleading approach. Using precision/recall or F1 score metrics instead will reveal the success of the developed system more clearly.

Rule-based systems were the initial ones developed to prevent fraud cases and these systems are still in use in several companies. In such systems, there are stages of creation of the rules, testing them, and making decisions by examining the cases that are not covered by the rules. All three stages are carried out manually. Therefore, rule-based systems have semi-automated traditional mechanisms where manual effort is high and complex patterns are difficult to detect. Some studies have been conducted in order to reduce the manual effort in rule creation and management processes and to increase the success rate. For example, Garcia [23] created a fraud-detection rule ontology using Web Ontology Language (OWL) and Semantic Web Rule Language (SWRL). In this study, the authors tried to strengthen existing rule-based systems with a semantic approach. They made use of a dataset containing anti-fraud rules of the existing rule-based system. First, fraud-detection rules and their connections were created. Afterwards, these rules were tested on a real fraud dataset and conflicts were revealed. These revealed conflicts showed the weak points of the system and these weaknesses were eliminated. As a result of this study, it was observed that the accuracy of the results increases by eliminating the rules that lead to wrong decisions. In another rule-based fraud-detection study, Febriyanti [24] aimed at eliminating the negative effect of strict rules on success by generating dynamic rules as a result of the analysis of business processes. In this study, a rule-based system that can detect fraud by adapting to rapidly changing event logs in the ERP system is proposed. As a work subject, the land-management process in the ERP system of a sugar company is discussed. They utilized a dataset that consists of event logs of the business process. These log records have a data model determined according to business processes. Instead of creating static rules, a mechanism was developed that could extract dynamic rules from business processes. The accuracy increased from 74% to 96% after these automatically generated dynamic rules were combined with the existing static rules. Although Garcia [23] uses methods that will increase the success of rule-based systems in his work, it requires manual effort to create and control these rules. In addition, in these studies, fraud detection was performed with narrow rules, making it difficult to reveal complex patterns.

Fraud detection is becoming more complex with each passing day, especially due to many zero-day attack techniques. Fraud characteristics do not remain constant and are in a continuous evolution process. For these reasons, fraud-detection mechanisms need to dynamically renew themselves and adapt to unprecedented fraud methods. However, this flexibility and adaptation is not possible via rule-based systems. Adding new rules and reorganizing existing rules according to changing fraud trends requires a great deal of manual effort. In addition, this process, which mostly proceeds with human effort, is open to errors. At this point, automated fraud-detection systems with high adaptability are favored. These systems have the ability to proactively defend against changing fraud flows with minimal manual intervention. While there may be various automated system architectures, machine learning and deep learning methods are mostly utilized in such systems. Alarfaj [8] compared the fraud-detection success of some machine learning algorithms and the CNN network in his study. The dataset consisted of two-day credit card transactions that took place in October 2018. The well-known shallow learning algorithms, including decision tree, k-NN, logistic regression, SVM, random forest, and XG Boost, were evaluated and compared against the CNN algorithm. The test results showed that the

structure they built using CNN had higher success on the basis of accuracy. However, accuracy is a misleading performance metric, since fraud detection has a high level of imbalanced datasets, whereas the F1 score parameter is one of the most accurate success metrics in the fraud detection field. In addition, because the study only covers credit card transactions for two days, it is important to note that the results for such short periods could be deceptive due to the nature of the fraud problem, since transactions in the fraud dataset may be related to transactions that took place a long time ago. Using a temporal distance in the training set as short as 2 days will prevent many fraud patterns from being revealed. Similarly, the results of the deep network architecture created in Yu's study [25] were compared with the results of some machine learning algorithms. It has been claimed that the results obtained with deep learning techniques are more successful. However, in this study, the measurement results were evaluated through the accuracy metric and there was no cost-based measurement system. In addition, no solution has been found to the problem of ignoring the connected history caused by the nature of transactional datasets. On the other hand, Chang [26] claimed that the selection of features that affect the decision mechanism in fraud detection is the most critical process. They claimed that since internet finance scenarios and fraud characteristics are constantly changing, artificial feature production methods are insufficient and become time-consuming. In order to prevent this negation, they proposed an automated feature-engineering method based on deep feature synthesis and feature selection. In this method, feature production is carried out by dividing the original data table into small pieces. The efficiency of the process of creating features and selecting important features for fraud detection is increased. This method does not require the opinions of business experts and significantly reduces manual effort during the feature-engineering phase. However, if a balanced model cannot be created with precise auto-tuning during the feature-engineering phase, overfitting problems will arise. Thus, the features produced will only be successful in the training data. In addition, the deep feature synthesis method may cause high operating costs on large datasets.

Since fraud is a temporal and historical problem, having information about past records is one of the most important requirements for detection systems. Therefore, we claim that the success achieved using only the transactional dataset cannot exceed a certain level. Different methods including transforming the dataset to include historical connections or using algorithmic memory techniques can be applied to gain this past awareness capability in the fraud-detection system. Few studies have utilized this idea. Olszewski et al. [27] tried to create a map of users using the self-organizing map (SOM) technique in order to build more successful fraud-detection systems. During this study, not vectors as in the original form of the SOM technique, but matrices consisting of records were included. Thus, the consecutive activities of the users are recorded in these matrices. While trying to detect fraud using these matrices, a new threshold mechanism is proposed. In this study, a grouping was made only on the basis of users' activities. It is aimed to detect fraudulent transactions based on the anomaly of the user's activities. However, objects with their own activity history, such as credit cards, were ignored. Malekian [28] proposed a new mechanism for profile-based fraud-detection systems that could adapt themselves to changing fraud patterns. In this proposed approach, besides the historical profile, there is also a temporary profile where new concepts can be perceived. Both profiles are fed with transactions in a real-time flow. It is understood that at the point where the temporal profile and the historical profile make different decisions, new concepts emerge and the current model begins to lose its up-to-dateness. One of the shortcomings of this study is that only changes in cardholder behavior were taken into account. If attempts are made with more than one credit card, the system will not be able to detect any changes. In order to prevent this problem, changes in all data objects that may affect the result must be evaluated. Moreover, Seyedhossein [29] claimed that a transaction-level fraud-detection system cannot achieve its maximum potential success. Instead, they proposed a profile-based fraud-detection system where profiling was performed on the daily amounts spent for each credit card and used in the decision phase. They tried to reveal the patterns through the changes in daily

spent amounts and the movements over time. In this way, they tried to shorten the time elapsed between the realization and detection of the fraud attempt. However, in this study, only credit card-specific profiling was performed and only the statistics of daily spending amounts were recorded. This is one of the earliest studies exploiting profiling mechanisms but far from multi-modal profiling and/or benefiting from rich profile details.

One of the approaches in which past transactions are taken into account is deep network architecture created via LSTM, where the significant information is kept in memory units. There are a few studies using LSTM to ensure the connection and continuity between transactions in the transactional dataset, such as [30–33]. However, where no clustering or profiling is performed on the data in studies using LSTM in this way, the memory units to be created will have a global scope, so unrelated transactions will also be considered. This will cause information complexity in terms of historical connection and the system will not be fully history-aware. In addition, LSTM would lack some profiles that may affect the result and many statistical data fields belonging to them. Thus, it could not provide this level of historical connectivity capability.

To the best of our knowledge, the fraud-detection problem in the aviation industry is hardly discussed and no comprehensive results have been presented so far. This study is the first in the field of aviation to address the fraud-detection problem, to try to develop solutions against challenges, and to present the results comprehensively. In this study, we first give the baseline, then we select the appropriate instances using the proposed BRUS algorithm from the whole dataset in order to remove the negative effect of random selection. We additionally benefit from the oversampling technique and combine it with the BRUS algorithm to find the optimum ratio between fraud and non-fraud instances for a robust fraud-detection model in aviation. We then introduce novice-engineered features obtained from the proposed profiles using the historical connection of passenger activities. Finally, we evaluate the performance of our mechanism not just using classical metrics but also including cost-sensitive measurements.

## 3. Dataset

In this study, we worked on an airline payment dataset that contains 61 features including 60 categorical fields and one numeric field. There are 37,303,697 transactions collected between October 2017 and December 2018. The detailed map of available features for both classification and profiling is given in Table 1. Airline activities are composed of 5 basic groups including Flight Data, Passenger Data, Cardholder Data, Payment Data, and Auxiliary Data, whereas banking datasets contain only common detailed credit card transactional features. Fraud datasets in aviation are formed by marking fraud records with the information of chargebacks coming from relevant banks. Thus, determining the positive records is a delayed process that is dependent on the credit card owners' complaints. In this dataset, there are 5512 fraud transactions; hence, the fraud ratio is 0.02%. On the other side, our dataset has 18.29% missing values overall, since some data fields are optional during the purchasing operation. In addition to this, an airline company has different sales channels and these channels have their own different processes. Since fraud is an evolving problem, a fraud-detection system needs to handle payment transactions in a timewise flow. Otherwise, the system will recognize future transactions, which is contrary to real life, and the success results will be misleadingly high. Due to these reasons, firstly, the transactions in the dataset were sorted chronologically.

**Table 1.** Available features in our dataset supported by a worldwide airline company.

| Feature | Group | Status |
| --- | --- | --- |
| Request_Date | Flight Data | Profiled and Discarded |
| Payment_Instrument | Payment Data | Used |
| Office_Number | Auxiliary Data | Used |
| Amount | Payment Data | Used |
| Currency | Payment Data | Used |
| TDS | Payment Data | Used |
| MD_Status | Payment Data | Used |
| Installment | Payment Data | Used |
| Payment_Service_Provider | Payment Data | Used |
| SubChannel | Payment Data | Used |
| Sale_Result_Code | Payment Data | Used |
| Client_OS | Passenger Data | Used |
| CardHolder_Name | Payment Data | Profiled and Discarded |
| CardHolder_Surname | Payment Data | Profiled and Discarded |
| Email | Passenger Data | Profiled |
| Phone_Number | Passenger Data | Profiled and Discarded |
| FFP_Number | Passenger Data | Used |
| Card_Number | Payment Data | Profiled and Discarded |
| Card_Origin_Country | Payment Data | Used |
| Card_Brand | Payment Data | Used |
| Card_Program | Payment Data | Used |
| Card_Type | Payment Data | Used |
| Origin_Country | Flight Data | Used |
| Destination_Country | Flight Data | Used |
| Reservation_Code | Flight Data | Used |
| Journey_Type | Flight Data | Used |
| Flight_Type | Flight Data | Used |
| Departure | Flight Data | Used |
| Arrival | Flight Data | Used |
| GMT | Flight Data | Used |
| Flight_Date | Flight Data | Profiled and Discarded |
| Flight_Time | Flight Data | Profiled and Discarded |
| Carrier | Flight Data | Used |
| Cabin | Flight Data | Used |
| Cabin_Class | Flight Data | Used |
| Passenger_Name | Passenger Data | Profiled and Discarded |
| Passenger_Surname | Passenger Data | Profiled and Discarded |
| Response_Value | Payment Data | Used |
| Client_Name | Auxiliary Data | Used |
| Charged | N/A | Class |

## 4. Challenges

Some common challenges are faced in most fraud-detection studies in the literature. All of the challenges mentioned in this section are also encountered in the aviation field. Since the number of transactions in the aviation field has reached very high volumes in the last period, the problems related to cost and large datasets can be especially devastating.

### 4.1. Imbalanced Dataset

Non-fraud records are far more frequent than fraud ones in almost all fraud-detection domains. Therefore, this problem is encountered in almost all fraud-detection studies in the literature [20,34,35]. Aviation is one of these domains that has a high level of dataset imbalance in their fraud-detection processes. Due to this problem, fraud-detection systems tend to make negative decisions (prone to non-fraud decisions) to acquire higher accuracy values. However, this fact will result in lower recall and F1 scores, which means that the fraud-detection system cannot detect fraud transactions properly. Sampling techniques are the most common ways to solve this problem:

- In the undersampling method, all fraud records are preserved and some of the legitimate records are selected according to a target fraud/legitimate balance. This target balance can be determined depending on the domain, interconnected transaction density, and dataset structure. Also, legitimate records can be selected randomly, homogeneously, or with a custom algorithm that decides considering the links between transactions in a smart way [36,37].
- In the oversampling method, all legitimate records are preserved and some synthetic fraud records are produced in order to achieve the target fraud/legitimate balance. These synthetic fraud records can be duplicates of current fraud records or completely new records based on the values of fraud records [38,39].
- Hybrid sampling methods are commonly used when undersampling methods are ineffective but oversampling methods are not possible to apply because the dataset is too big to reproduce new records [22,40,41]. In this approach, the undersampling method is used to some extent; thereafter, oversampling is applied to selected legitimate records. In other words, there is a two-phased sampling process in hybrid sampling methods.

### 4.2. Smart Instance Selection

This is a sub-problem of the imbalanced dataset problem that shows up while using undersampling methodology to overcome the main problem. Undersampling is one of the most basic methods applied to solve the imbalanced dataset problem. However, a sampling approach that uses a completely random selection technique can result in the selection of unrelated records. This may adversely affect the fraud-detection process and lead to a decrease in success scores. For this reason, a smart instance selection mechanism that selects legitimate records with a custom smart algorithm can achieve more successful scores instead of a random manner.

### 4.3. Cost-sensitive Measurement

Fraud is often a financial problem, as it directly affects companies' potential profits and losses. Therefore, the quantity-based success criterion can cause monetary losses to companies in detecting fraud. However, most of the studies [42–44], except for a few [45,46] in the literature, work on fraud-detection systems that measure success based on quantity. Serious financial losses are experienced in the airline industry due to fraudulent transactions. Therefore, the primary purpose of a fraud-detection system to be developed in this area should be to reduce fraud-related losses and prevent potential decreases in income. In this research, a cost-centric approach to measuring success was also considered, with the primary objective being the detection of fraud and the minimization of financial losses for airline companies.

Cost-sensitive measurement brings forward the possible negative impact of the quantity-based evaluation and possible revenue losses that arise from the identification strategy of a fraud-detection system. Measuring the success of fraud-detection systems in terms of quantity-based metrics is completely against the financial nature of fraud detection. Thus, cost-sensitive metrics and a cost-based perspective are necessary for fraud-detection systems to observe and compare the cost- and quantity-based approaches.

*4.4. Fraud Detection on Transactional Datasets*

Many studies published about fraud detection have tried to detect fraud in datasets consisting of credit card transactions [47–49]. However, since fraud is a temporal and constantly evolving problem, the past links of the transaction to be decided are also of great importance. Due to the absence of historical connections within datasets comprising transactional records, a substantial amount of information pertaining to the relevant transaction may prove to be inadequate during the fraud-detection phase. This can lead to the failure to detect a potentially fraudulent transaction, or vice versa, to the false detection of a reliable transaction. In order to solve this problem, it is important that the dataset is chronologically ordered within itself and between the training and test sets. Then, the dataset should be made historically sensitive or a retrospective control/memory mechanism should be developed at the algorithmic level.

In the aviation industry, the interconnection of transactional data is richer than simple credit card transactions. Thus, some profiles could be created exploiting this interconnection to build a history-aware fraud-detection system and to overcome the transactional dataset challenge in the aviation fraud area. To the best of our knowledge, fraud-detection systems utilizing historical information have not been developed in the aviation field. In this study, we propose a profile-based fraud-detection system that consists of aviation-specific profiles to solve this common problem.

## 5. Profile-Based Fraud Detection

Since fraud detection is a chronological and temporal problem, the past movements of the relevant transactions are also of great importance. However, several studies [50–53] utilize only transaction information, ignoring their relation to the past. This paradigm causes fraud-detection systems only to have information about the relevant transaction at the time of the decision and completely ignore the past activities of respective user/card/PNR (Passenger Name Record) information. This common approach, which is against the nature of fraud detection, limits the success rates of those systems by failing to notice high-risk user and payment information and/or rejecting ordinary safe ticket sale activities. To address this issue, it is essential to leverage historical transactions by constructing meaningful airline passenger profiles. These profiles can assist in recognizing trustworthy user and card activities, allowing them to be directly integrated into the decision-making process.

In this study, an ability to be aware of the past movements of candidate passengers has been enabled by deriving them from historical statistics of certain airline customer profiles and adding them to the current transactional information. The proposed profile-based fraud-detection system for aviation is built on three basic profiles, as follows:

- Credit Card Profile
- PNR Profile
- Record Profile

Although both the PNR profile and the credit card profile are basically similar profiles, they are separated because they are quite different from each other in terms of their scope. The scope of the credit card profile covers the entire dataset, whereas the PNR profile covers a limited area such as all attempts, additions, and cancellations made by the customer from entering into the system and starting to request a service until the end of this service. The credit card and PNR profiles created in this study benefit from the historical movement strategy. In such profiles, the process is progressing by generating new data fields, which consist of statistical data that are calculated by considering all relevant past transactions



of the profile value. In order to utilize such historical movements, first, records must be processed chronologically, just as real-world transaction flow. Thus, the profile information in each transaction only represents the previous history of that transaction. In this way, each record in the dataset will have information not only about themselves but also about all previous transactions that have been linked to them. In contrast, the record profile exploits a local approach to reveal some hidden information in the record. This profile is designed to reveal hidden information in the record and to generate new data fields that have a direct impact on decision-making. On the other hand, some profiling modes have been designed so that the created profiles can be used in different combinations. Using these profiling modes, it will be possible to observe which profiles have more effect on the result. These profiling modes will be explained in detail in the methodology section.

In the following subsections, we aim to illustrate the effectiveness of the proposed profiles by presenting several example scenarios using Tables 2–4. It is important to note that the "chargeback" column serves as the ground truth indicator for non-fraud and fraud cases. Additionally, all email addresses and passenger names shown in Tables 3 and 4 are fictional and conceived by the authors just to demonstrate the context.

Another important issue is that there may be sudden changes in spending habits in extraordinary situations such as COVID-19. However, since sudden changes in such situations will occur in all profile objects at the same time, they can be distinguished from fraud cases. Spending changes in fraud cases are mostly more drastic. On the other hand, fraud attacks are evolving in time, so fraud models need to be frequently updated accordingly. Hence, our profile mechanism would adapt itself those changes.

**Table 2.** Two example credit card profiling scenarios to emphasize the difference in the capability of fraud detection between transactional and profile-based approaches.

| Credit Card Profile | | | | | | | |
|---|---|---|---|---|---|---|---|
| **Transactional Fields** | | | | | **Profile Map** | | |
| **Transaction** | **Fraud Attempt** | **Approval** | **Chargeback** | **Amount** | **Approval Count** | **Chargeback Count** | **Average Expense Amount** |
| **Scenario 1** | | | | | | | |
| T1 | No | Accept | No | $600 | 0 | 0 | $0 |
| T2 | No | Accept | No | $300 | 1 | 0 | $600 |
| T3 | No | Accept | No | $500 | 2 | 0 | $450 |
| T4 | No | Accept | No | $600 | 3 | 0 | $466.67 |
| T5 | Yes | Accept | Yes | $3000 | 4 | 0 | $500 |
| T6 | Yes | Reject | No | $3500 | 5 | 1 | $500 |
| T7 | Yes | Reject | No | $2500 | 5 | 1 | $500 |
| **Scenario 2** | | | | | | | |
| T1 | No | Accept | No | $150 | 0 | 0 | $0 |
| T2 | No | Accept | No | $200 | 1 | 0 | $150 |
| T3 | No | Accept | No | $75 | 2 | 0 | $175 |
| T4 | No | Accept | No | $125 | 3 | 0 | $108.33 |
| T5 | No | Accept | No | $250 | 4 | 0 | $112.5 |
| T6 | Yes | Accept | Yes | $2000 | 5 | 0 | $140 |
| T7 | Yes | Reject | No | $2500 | 5 | 1 | $140 |

**Table 3.** Two example PNR profiling scenarios to emphasize the difference in the capability of fraud detection between transactional and profile-based approaches.

| | | | PNR Profile | | | | | | | |
|---|---|---|---|---|---|---|---|---|---|---|
| | Transactional Fields | | | | | Profile Map | | | | |
| Transaction | Fraud Attempt | Approval | Chargeback | Email | Phone Number | Accept Count | Reject Count | Chargeback Count | Unique Email Count | Used Phone Numbers |
| Scenario 1 | | | | | | | | | | |
| T1 | Yes | Reject | No | richard4355@big.co | 0505 | 0 | 0 | 0 | 0 | 0 |
| T2 | Yes | Reject | No | andy77@anyco.com | 0532 | 0 | 1 | 0 | 1 | 1 |
| T3 | Yes | Reject | No | john4@company.com | 0555 | 0 | 2 | 0 | 2 | 2 |
| T4 | Yes | Reject | No | john5@company.com | 0534 | 0 | 3 | 0 | 3 | 3 |
| T5 | Yes | Accept | Yes | safemail@safe.com | 0535 | 0 | 4 | 0 | 4 | 4 |
| T6 | Yes | Reject | No | safemail@safe.com | 0535 | 1 | 4 | 1 | 5 | 5 |
| T7 | Yes | Reject | No | gerard123@fake.com | 0533 | 1 | 5 | 1 | 5 | 5 |
| Scenario 2 | | | | | | | | | | |
| T1 | Yes | Reject | No | suspicious1@mail.com | 0531 | 0 | 0 | 0 | 0 | 0 |
| T2 | Yes | Reject | No | suspicious2@mail.com | 0532 | 0 | 1 | 0 | 1 | 1 |
| T3 | Yes | Reject | No | suspicious3@mail.com | 0533 | 0 | 2 | 0 | 2 | 2 |
| T4 | Yes | Accept | Yes | deceptive1@mail.com | 0534 | 0 | 3 | 0 | 3 | 3 |
| T5 | Yes | Reject | No | deceptive1@mail.com | 0534 | 1 | 3 | 1 | 4 | 4 |

**Table 4.** Nine independent record profiling scenarios to emphasize the difference in the capability of fraud detection between transactional and profile-based approaches.

| | | | Record Profile | | | | | | |
|---|---|---|---|---|---|---|---|---|---|
| | Transactional Fields | | | | | | | Profile Map | |
| Transaction | Fraud Attempt | Approval | Chargeback | Transaction Date | Flight Date | Credit Card Owner | Passengers | Remaining Minutes To Flight | Passenger Surname Matched |
| T1 | No | Accept | No | 21/12/2022—11:30 | 28/12/2022—19:00 | Anthony Kent | Julia Kent | 10,530 | Yes |
| T2 | No | Accept | No | 15/02/2023—10:42 | 28/06/2023—19:00 | Thomas Johnson | Madelyn Johnson | 191,718 | Yes |
| T3 | No | Accept | No | 06/08/2022—22:24 | 02/09/2022—06:00 | Bill Morris | Bill Morris, Alice Morris | 37,896 | Yes |
| T4 | Yes | Accept | No | 28/12/2022—21:30 | 28/12/2022—23:20 | Cris Jackman | Tony Simmons, Andrei Fox | 110 | No |
| T5 | No | Accept | No | 10/02/2023—14:22 | 02/03/2023—07:15 | Leonie Larson | Rosanna Larson | 28,373 | Yes |
| T6 | No | Accept | No | 28/09/2022—08:56 | 14/11/2022—10:30 | Marianne Wallace | Danielle Wallace | 67,774 | Yes |
| T7 | No | Accept | No | 07/05/2023—11:43 | 25/05/2023—14:55 | Findlay Abbott | Findlay Abbott, Valentina Abbott | 25,728 | Yes |
| T8 | No | Accept | No | 25/12/2022—22:24 | 01/01/2023—04:30 | Jensen Erickson | Gavin Rose, Lukas Erickson | 9006 | Yes |
| T9 | Yes | Accept | No | 11/07/2023—02:38 | 11/07/2023—04:15 | Laura Holland | Milan Molina, Adrian Brennan | 97 | No |

## 5.1. Credit Card Profiling

This profiling mechanism focuses on the past activities of a given credit card. It scans the entire history of the respective card, as given in Table 2. Thus, we collect feedback about the card used for purchasing an airline ticket and we form an opinion about the reliability of this card. The following fields are extracted as novice features from the past transactions of the given card:

- Number of chargebacks in the past
- Number of accepted transactions in the past
- Number of rejected transactions in the past
- Number of transactions submitted for manual review in the past
- Number of domestic flights purchased with this credit card
- Number of international flights purchased with this credit card
- Number of unique phone numbers used with this credit card
- Number of unique emails used with this credit card

- Average amount spent with this credit card
- Highest amount spent with this credit card

Table 2 contains two credit card profiling scenarios that present the status of the profiling map when each transaction arrives. These scenarios represent how cases that cannot be detected from a transactional perspective can be caught using changes in spending habits from a profile-based perspective. While colorless regions represent legal transactions, green regions represent fraudulent transactions that can be detected from the proposed perspective, and red regions represent fraudulent transactions that cannot be detected from the transactional perspective. Substantial rises in expenditures were observed in transactions marked in red. These increases might show that the card has been taken over by a fraudster and fraudulent transactions are being attempted. However, in transactional datasets, these relative changes cannot be caught by the fraud-detection system. One of the main goals of our credit card profile is to be aware of the spending habits of related credit cards. Thus, a fraud-detection system will be capable of perceiving the suspicious differences between current transactions and past statistics.

The card has been stolen and used by fraudsters in red-marked transactions. Some fraudsters try to purchase high-priced tickets in order to gain the maximum profit. Therefore, after the card is stolen, the spending habits related to the card may change suddenly. Although it is not possible to capture this change with the transactional approach, it can be easily detected with the profile-based approach. With the transactional detection system, the red-marked transaction cannot be detected and then a chargeback occurs. However, it is later understood that this card has been stolen and the green-marked transactions under the red-marked transaction can be blocked even though they are fraud. In summary, the transactional approach will miss all fraud transactions of a stolen card from the moment the first fraudulent transaction occurs until the chargeback feedback is received. However, since the profile-based approach can take into account transaction histories, statistical information, and changes, it has the ability to catch all fraudulent transactions from the first attempt.

### 5.2. PNR Profiling

PNR refers to a session that covers all transactions of the user after interacting with the sales channel until the end of the customer experience. In this profiling mechanism, we aim to track the short-term activities of a customer regarding the related session. Our analysis of the fraud cases shows us that fraudsters' activities generally differ from ordinary customers. The following features are engineered via a given PNR number:

- Number of chargebacks in the past
- Number of accepted transactions in the past
- Number of rejected transactions in the past
- Number of transactions submitted for manual review in the past
- Number of domestic flights purchased in this PNR
- Number of international flights purchased in this PNR
- Number of unique credit card numbers used in this PNR
- Number of unique phone numbers used in this PNR
- Number of unique emails used in this PNR

It is obvious that purchase attempts have been made with more than one e-mail and phone number information in the same PNR of two different scenarios in Table 3. In such cases, the transactional perspective proves ineffective in managing PNR transactions comprehensively, causing it to fail. Conversely, the profile-based approach excels at identifying fraudulent activities. The green regions indicate where the proposed perspective can successfully detect fraudulent transactions, while the red regions signify fraudulent transactions that remain undetected when using the transactional perspective. The fraud-detection system rejected the first attempts because the given e-mail or phone number information poses a high risk. However, the system made a wrong decision on the red attempt by

accepting the transaction, and the fee was refunded due to the fraud report. When the transactions are examined visually, it is very clear that we are dealing with a fraudster trying to circumvent the system. Nevertheless, since there is no connection between these transactions in the transactional dataset, this suspicious activity cannot be detected. In order to prevent such vulnerabilities, a PNR profile was created and statistical information for all trials performed on the relevant PNR was added to the records. In this way, a record is made aware of all attempts in the PNR to which it belongs.

### 5.3. Record Profiling

In this profiling mechanism, new distinctive features are extracted from multiple columns placed only within the record itself. The following two features helped us to detect several suspicious activities during ticket sales.

- The number of minutes remaining on the flight at the time of the transaction
- Whether the surname of the credit card holder matches the surname of any passenger

In Table 4 there are nine individual record profiling scenarios to represent how some risky patterns can be detected by making the date information and name/surname alignment information in the record meaningful. By comparing the cardholder and passenger information and calculating the time remaining until the flight, cases that go against the natural process can be detected. While colorless regions represent legitimate transactions, green regions represent fraudulent transactions that can be detected from the proposed perspective, and red regions represent fraudulent transactions that cannot be detected from the transactional perspective. In legitimate transactions, the submitted and temporal information seems proper. However, in fraudulent transactions, there are some inconsistencies and cases that may pose a risk in the record. First of all, the fraudulent transactions took place a short time before the flight. This event can be used as a fraud strategy so that there is no time left for manual examination of suspicious transactions. Moreover, in fraudulent transactions, the surname of the cardholder and any of the surnames of the passengers do not match. While this situation may occur naturally when a ticket is purchased for someone outside the family, it also occurs in every fraud attempt. Manual reviews can easily uncover these two suspicious cases. However, due to the dispersion of meaningful information across multiple areas, the system may struggle to detect them efficiently. Hence, within the record, the process of extracting meaningful information that is distributed across multiple fields and converting it into new fields was executed during the record profiling stage.

## 6. Methodology

The system architecture of the developed fraud detection system, created with a holistic view, is given in Figure 1.

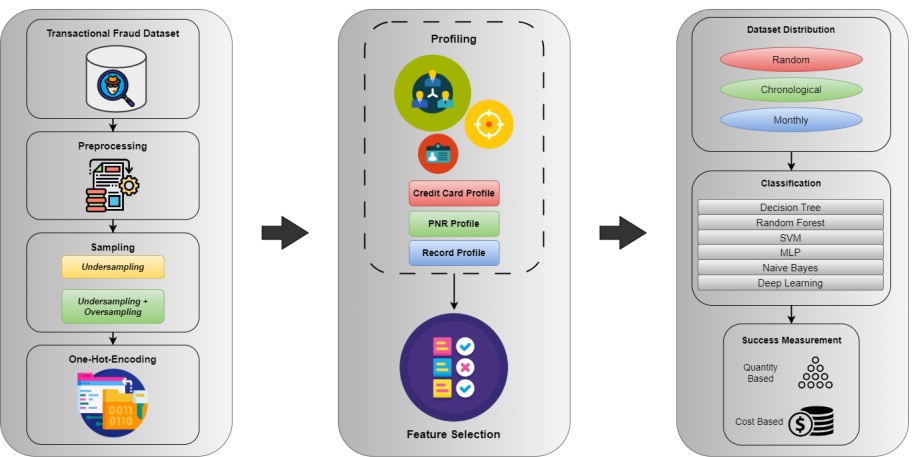

**Figure 1.** The system overview of the fraud-detection system for aviation.

### 6.1. Data Pre-Processing

At this stage, various operations were applied to the raw data in a single chronologically ordered text file in order to eliminate the data fields with high numbers of missing values and to correct the data containing misleading values. In total, 4369 records and 32 data fields were removed from the dataset for the following reasons.

- Non-standard and incorrect representations of date fields
- Lines that have lost their structural feature and are corrupted
- Data columns with a high rate of missing values

We also filled in the missing values in the categorical data fields that could be inferred according to some business logic. In this filling process, mapping methods were used based on similar records, and assumptions were made according to some aviation business processes. Other steps applied in the data pre-processing phase are given as follows:

- Unexpected tokens in the records were corrected or removed from the dataset.
- Data values of the same type were standardized to have the same representation.
- Columns that have dirty data and are ineffective were removed from the dataset.
- Type checking and correction were applied to numeric values.
- Data fields with a high number of categories were excluded from the dataset.
- Different currencies in the dataset were converted to US dollars.
- Min–max normalization was applied to numerical fields.

After the cleaning operations, a dataset consisting of 37,299,328 transactions and 29 data fields with only one numerical field was obtained.

### 6.2. Sampling

Fraud datasets have a very low percentage of fraud records, causing a serious imbalance problem. The most-commonly used approach for solving this problem is to create a new dataset using sampling techniques on the existing dataset [31]. Decreasing the number of majority records by randomly or smartly selecting them to some degree while all positive records are included in the new dataset is called undersampling. Similarly, the method of synthetically increasing the number of positive records in the dataset is called oversampling [32]. However, while working on a very large dataset, as we used in our study, it may not be possible to directly apply the oversampling method. In such cases, hybrid approaches can be used where undersampling and then oversampling are utilized in a cascaded manner. Two different sampling methods were used in this study. The first one is the balanced random undersampling (BRUS) method, and the second one is the hybrid BRUS + oversampling method.

#### 6.2.1. Balanced Random Undersampling (BRUS)

The balanced random undersampling method basically aims at increasing the fraud/legitimate ratio by making a choice between legitimate records. However, it is different from the random undersampling methods that are frequently used in the literature and library implementations. These qualities are as follows:

- ✓ It is guaranteed that the data to be selected from the majority class will be evenly distributed in the dataset
- ✓ Chronological order is guaranteed in the newly created dataset
- ✓ It is not possible to select some records more than once while there are records that have not been selected yet

In order to guarantee these principles and for the BRUS algorithm to work correctly, the dataset must be ordered according to the transaction times. First of all, by comparing the fraud/legal rate in the dataset and the rate desired to be reached, it is decided to what extent the legal records should be reduced. In order to make a balanced negative sample selection on the dataset, the dataset is divided into bins according to the ratio obtained in the previous step. A random negative sample is selected for each created bin,

and the desired fraud/legal ratio is reached. Thus, highly representative records with a well-balanced spread over the dataset are selected, and duplication is prevented. Therefore, the conditional differences between different tests on the same dataset will be minimized. The fundamental steps of BRUS are presented in Algorithm 1.

---

**Algorithm 1** BRUS Algorithm

---

**DETERMINATION OF BIN INTERVAL**

1: $totalCount, fraudCount, binInterval \leftarrow 0, 0, 0$
2: **for each** $passengerActivityRecord \in dataset$ **do**
3: 　　**if** $passengerActivityRecord$ $is$ $fraud$ **then**
4: 　　　　$fraudCount \leftarrow fraudCount + 1$
5: 　　**end if**
6: 　　$totalCount \leftarrow totalCount + 1$
7: **end for**
8: $targetLegalCount \leftarrow fraudCount * (targetLegalRatio / targetFraudRatio)$
9: $binInterval \leftarrow totalCount / targetLegalCount$

**NEGATIVE SAMPLE SELECTION**

10: $counter, selectIndex \leftarrow 0, 0$
11: **for each** $passengerActivityRecord \in dataset$ **do**
12: 　　**if** $counter$ % $binInterval = 0$ **then**
13: 　　　　$selectIndex \leftarrow random.nextInt(downRatio)$
14: 　　**end if**
15: 　　**if** $counter$ % $binInterval = selectIndex$ **then**
16: 　　　　**if** $passengerActivityRecord$ $is$ $not$ $fraud$ **then**
17: 　　　　　　$select(passengerActivityRecord)$
18: 　　　　**else**
19: 　　　　　　$selectIndex \leftarrow selectIndex + 1$
20: 　　　　**end if**
21: 　　**end if**
22: 　　$counter \leftarrow counter + 1$
23: **end for**

---

Since the majority of negative records are not selected in undersampling methods, the continuity of interconnected records may be lost. In order to eliminate this problem, the entire dataset was profiled before sampling was applied. Thus, a dataset that is aware of previous related records, not only on a user basis but also on a credit card and PNR basis, has emerged. After the profiling stage, statistical information on past transactions is added to all transactions in the dataset. In this way, the records selected with the BRUS algorithm were connected with all unselected related records through profiles.

### 6.2.2. Oversampling

Oversampling is a sampling method that is generally used to prevent data loss in negative instances. There are several oversampling algorithms like random oversampling, SMOTE, and ADASYN. However, when an oversampling method is directly used on the dataset, the number of total instances will be higher than the original dataset. Because of this fact, a direct oversampling method was not chosen in our study, as we have a very large dataset.

### 6.2.3. Hybrid BRUS + Oversampling

In cases where the studied dataset contains a large number of records, applying oversampling methods directly may cause resource insufficiency or performance problems at other stages. In this case, the created model completely loses its applicability. In order to avoid this problem, first of all, by applying balanced random undersampling as developed in our study, the fraud/legal ratio was increased to 1:25 and 1:100, and then by

applying various oversampling methods on these intermediate data sets, it was reduced to a fraud/legal ratio of 1:5, which we used as the base ratio in our study. The utilized methods for the oversampling phase in the hybrid sampling mechanism are as follows:

- Random Oversampling: In this method, synthetic records are produced by randomly selecting and copying records from the minority class. It is important for the success of the detection system that the oversampling algorithm to be developed has a structure that can prevent unbalanced distribution and unbalanced selection.
- SMOTE: This is a well-known algorithm that is used in many fraud-detection studies to solve the imbalanced dataset problem [21,54,55]. Minority class records are selected to find k-nearest neighbors of these records. Afterward, a synthetic record is produced by selecting a point between the nearest neighbor and the selected record. Since this algorithm is not suitable for processing categorical data fields and there is a high number of categorical data fields in our data set, it has been observed that the success rates are generally very low.
- SMOTENC: This is a variation of the SMOTE algorithm that can also handle categorical data fields [56,57]. With the application of this technique, it has been observed that the low success observed in SMOTE is prevented and it exhibits more balanced results compared to the random oversampling technique.

The sampling step was applied before the one-hot-encoding process in order to keep the processing times short and to reduce the resource consumption in SMOTE and SMO-TENC algorithms. The dataset was made ready for the sampling step by applying only label encoding. The one-hot-encoding process was carried out in the next step.

### 6.3. Profiling

6.3.1. Profile-Based Transformation

In order to develop the profile-based fraud-detection system, the dataset should be converted into a profile-based form. Thus, the transactional records in the dataset will have historical information and will be aware of the previous linked records. The profile-based transformation module applied to the dataset was developed using the Java programming language. Since this profiling process must be performed in accordance with real-world scenarios, the dataset must be chronologically ordered. As mentioned in the Profile-based Fraud Detection section, since the credit card and PNR profiles are historical profiles, their conversion processes are carried out using maps containing statistical objects. There is a map for each profile and a statistical object for each unique profile value. The chronologically ordered dataset is scanned for each profile from the first record to the last record, and some basic steps are applied to each record. As a result of the completion of the scanning iteration, the relevant profile is created and added to the dataset. The core steps of profiling mechanism is presented in Algorithm 2.

---

**Algorithm 2** Profiling Algorithm

---

1: **for each** $passengerActivityRecord \in dataset$ **do**
2:     $statistics_i \leftarrow profileMap[profilingObjectId_i]$
3:     **if** $statistics_i = null$ **then**
4:         $initialize(statistics_i)$
5:     **else**
6:         $newStatistics \leftarrow recalculate(statistics_i)$
7:     **end if**
8:     $passengerActivityRecord[j] \leftarrow statistics_i[j]$
9:     $statistics_i \leftarrow newStatistics$
10: **end for**

---

When a scan iteration is completed on the dataset by applying these steps for each record, the targeted profile is created. As the newly generated fields are incorporated into the respective record during the scanning process, the dataset undergoes a transformation

into a profiled dataset once the profiling scans are completed. After the whole profiling process is completed, the dataset is ready for the sampling step.

### 6.3.2. Profiling Modes

The principles that show how the designed profiles are used in our study can be called profiling modes. By using the existing transactional dataset and new data fields added to the profiles, new datasets in different combinations were created. Our fundamental tests were carried out with these datasets and the efficient profiling modes were determined. The performance analysis was conducted on the following profiling modes:

1.  Credit Card Profiling Mode: Transactional Dataset + Credit Card Profile
2.  PNR Profiling Mode: Transactional Dataset + PNR Profile
3.  Record Profiling Mode: Transactional Dataset + Record Profile
4.  Only Profiling Mode: All Profiles without Transactional Dataset
5.  Multi-Modal Profiling: Transactional Dataset + All Profiles

### 6.4. One-hot Encoding

Categorical fields are not directly utilizable for machine learning algorithms and deep neural networks (DNN). Although it is ready to be processed structurally when numbers are given to each category by applying the label-encoding technique, there will be still a semantic inconsistency in the label values. At this point, the one-hot-encoding method in which a binary data field is produced for each category can be used. Thus, a direct connection between the categories and the class can be established. By applying the one-hot-encoding process to 28 categorical fields in the dataset, 1524 binary features were obtained. With the feature-selection stage to be applied to quite a lot of binary fields, the most distinctive categories in all encoded data fields were selected.

### 6.5. Feature Selection

In order to determine discriminative features and minimize the processing time, we applied feature-selection algorithms to the available dataset. In our study, feature selection was performed with 3 different techniques, including $CHI^2$, PCA, and AutoEncoder, on 1524 transactional binary features, 1 numerical transactional feature, and 21 additional numerical profiled features. Since the desired success scores could not be reached with the number of features above 100, higher-resolution tests were carried out under the number of 100 features. As a result of these tests, it was observed that the most successful results were obtained in the tests performed with the feature numbers between 20–75. Afterwards, the following feature-selection techniques were applied in this range as a standard. Also, some lower and higher features are included in our experiments so that we can control how the results change outside this range.

### 6.6. Classification

In the classification phase of our fraud-detection system for aviation, five state-of-the-art machine learning algorithms and DNN were exploited. These below-given machine learning algorithms were chosen among the methods that are frequently used and compared in fraud-detection studies in the literature [8,58–62].

*   Decision Tree
*   Random Forest
*   Support Vector Machines
*   Multilayer Perceptron
*   Naive Bayes

In addition to the mentioned machine learning algorithms, some experiments were also carried out using deep neural networks. We also conducted several experiments to find the best hyperparameters, including the number of hidden layers, neuron formation, epoch number, batch size, learning rate, and dropout parameters, in order to fine-tune the

fraud-detection system based on the aviation domain. The initial parameters were chosen regarding the study [4], which aimed at detecting credit card fraud.

### 6.7. Cost-based Measurement

From the point of view of companies, since fraud is mostly a financial problem, the success criteria based on the number of transactions may not always be consistent. When the developed fraud-detection systems are analyzed according to the number of correctly detected transactions, they may seem successful, but the financial loss faced by the company might be much greater. In many fraud-detection studies in the literature, success is calculated based on only the number of transactions, and the monetary costs of the transactions are ignored [63–65]. Therefore, in this study, besides the success criteria based on the number of transactions, cost-based success criteria were also utilized. Accuracy, precision, recall, and most importantly F1 score metrics, which are widely used in fraud detection, are calculated on a cost basis. The cost-based metrics we propose in this study are named C-Accuracy, C-Precision, C-Recall, and C-F1 Score.

$$c = currencyUSDRatio \times amount$$
$$cScore = (c_i - min(c))/(max(c) - min(c)) \tag{1}$$

While calculating these proposed metrics, first of all, the transaction amounts were standardized by converting them according to currency conversion rates. Afterward, min–max normalization was applied to the amount column within the 0–1 interval. Finally, while creating the cost-based confusion matrix, each transaction was weighted with its c-score and added to the matrix. All metrics calculated using a cost-based confusion matrix have become cost-sensitive. In order to create a cost-sensitive confusion matrix, the operations for calculating the c-score are given in Equation (1).

### 7. Experimental Results

In this section, we first give the details of our experimental dataset. We then share the initial results of state-of-the-art learning algorithms and demonstrate the performance contribution of feature selection and sampling techniques to transactional information in terms of quantity and cost. Afterward, we elaborate on the success of the introduced profiling mechanism. Finally, we give the test results of DNN using the combination of hyperparameters in order to demonstrate the performance improvement of the proposed system. All the experiments are carried out to measure the state-of-the-art performance metrics, including accuracy, precision, recall, and F1 score. However, we mostly benefit from the F1 score, since it is the fairest metric for imbalanced dataset problems. The dataset configurations of these experiments are given in Table 5.

Since the fraud-detection problem is an imbalanced dataset problem, the accuracy metric can be misleading for measuring real success. Such datasets contain a very high number of negative (legal) samples. Thus, the developed detection systems may tend to make negative decisions in all transactions. In this case, the accuracy results will be observed to be very high, but will not reflect the truth. Therefore, precision and recall results should be evaluated together instead of accuracy in fraud-detection studies. To balance their impact, the F1 score obtained by calculating precision and recall metrics together is utilized as the primary metric for such problems. On the other hand, the precision metric gives us information about the rate at which ordinary passengers are wrongly blocked, whereas the recall metric shows how much of the fraudulent transactions are caught. The F1 score value represents how successful the fraud detection and false alarm prevention function is in a holistic manner. For all these reasons, the basic success criterion in this study was the F1 score.

**Table 5.** The details of our dataset and train/test setup.

| Dataset Configuration | | Value |
|---|---|---|
| | Fraud/Legitimate Ratio | 1:5 |
| | Dataset Time Range | 15 Months |
| **BRUS** | Number of Fraud<br>Number of Legitimate | 5512<br>27,578 |
| **Oversampling × 5** | Number of Fraud<br>Number of Legitimate | 27,560<br>137,890 |
| **Oversampling × 20** | Number of Fraud<br>Number of Legitimate | 11,240<br>551,560 |
| **#Features** | Transactional dataset<br>CC Profile<br>PNR Profile<br>Record Profile | 41<br>10<br>9<br>2 |
| **Distribution** | Type<br>Train/Test Ratio | Chronological<br>70/30 |

### 7.1. Performance Improvements on Transactional Dataset

In our previous work [5], some tests and improvements were carried out in order to achieve maximum success on the transactional dataset. During these tests, several feature-selection mechanisms, measurement metrics, and train/test set organization methods were explored in order to improve fraud-detection performance for aviation. In this study, oversampling techniques and deep learning approaches were analyzed on the airline ticket sale dataset and the results are presented in this subsection. It has been observed that there are important limitations to increasing the success of a transaction-based fraud-detection system in order to prevent financial loss for airline companies and fulfill the sales experience of airline customers. Despite all the improvement and optimization efforts, the precision, recall, and especially the F1 score metrics could not reach acceptable levels either on the number of transactions or on the cost basis.

The success rates after the feature-selection process is applied are given in Figure 2. In addition, the newly proposed cost-based measurement metric results are presented in Figure 3. The first test results using shallow and deep learning algorithms after performing only the necessary cleaning operations on the dataset are given as initial results in these figures. Examining these figures shows us that no significant change was attained for the decision tree algorithm after the feature selection was applied, whereas the naive Bayes algorithm performs greatly after feature selection. The primary cause of the unexpectedly poor performance observed prior to the pre-processing step is the correlation between the provided features. The performance of this basic algorithm was significantly improved by both the feature-selection process and the inclusion of engineered features. Apart from the DT algorithm, we observed a noticeable increase between 2.1% and 48.7% in terms of the quantity-based F1 score. The cost-based evaluation shows that the quantity–cost success balance is preserved and the most successful algorithm is SVM. Based on all these observations, it can be understood that the feature-selection process makes a substantial contribution to the fraud-detection problem in the aviation domain. Even the poor success results obtained via the naive Bayes algorithm increased after feature selection was applied and came to the same success level with other algorithms. In order to observe the effect of the feature-selection process, some tests have been carried out using the $CHI^2$ and PCA algorithms. The most successful classification algorithm was SVM, whereas the most successful feature-selection method was $CHI^2$. The number of features that gave the highest success results on average for all algorithms was 55. Also, a significant increase of up to 48.7% was observed in the naive Bayes test results. The details of the test results are given in our previous study [5].

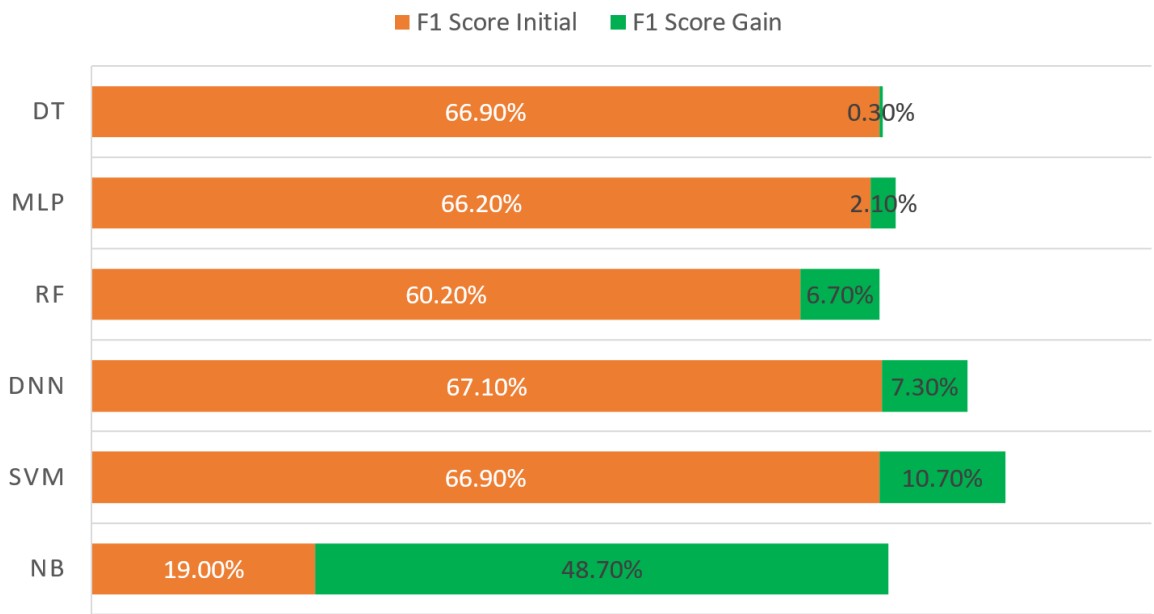

**Figure 2.** The initial fraud-detection performance of state-of-the-art learning algorithms and the impact of the feature-selection process.

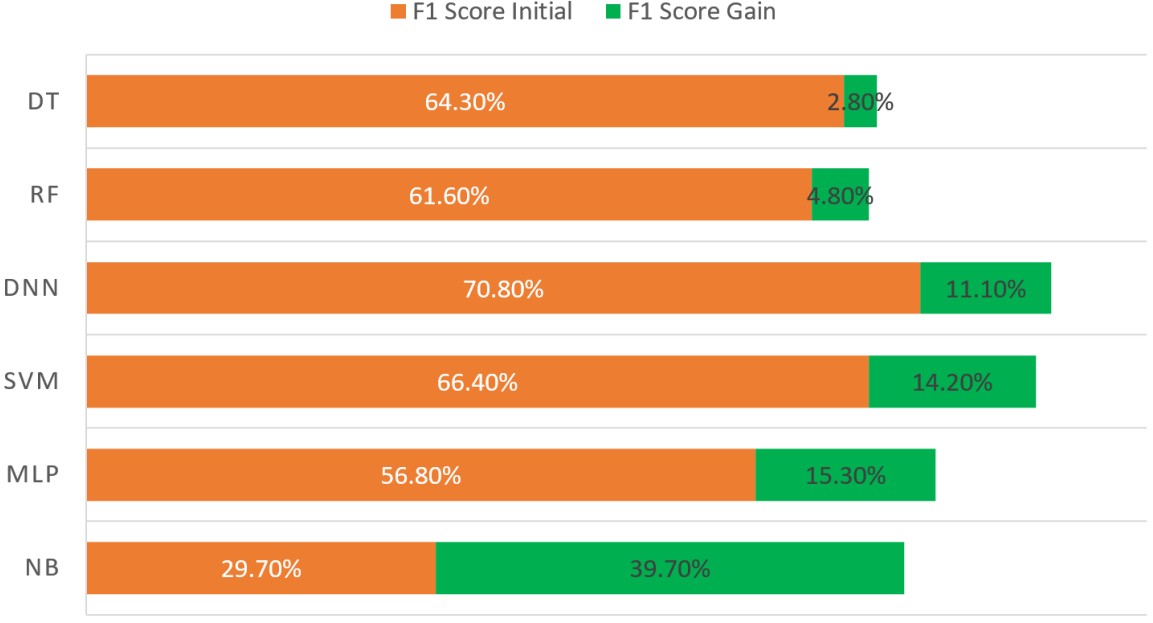

**Figure 3.** The initial fraud-detection performance of state-of-the-art learning algorithms and the impact of the feature-selection process in terms of cost.

Sampling is a crucial technique for improving the success of solving imbalanced dataset problems. In our previous sampling tests [5], we applied only the random undersampling method and we picked a 1:5 fraud/legal ratio as the most successful configuration regarding the test results. The outcomes of the hybrid oversampling methods we implemented to address the issue of imbalanced datasets, along with their comparison to the BRUS algorithm proposed in our study based on F1 score, are presented in Figures 4 and 5. Oversampling algorithms were applied in a hybrid manner together with the BRUS algorithm. First, very low success results were obtained since the SMOTE algorithm is not suitable for handling categorical fields. Therefore, the SMOTE algorithm

was excluded and its results are not included in this section. Comparing the SMOTENC and Random Oversampling algorithms showed that SMOTENC has a higher success rate of 20.6% and 10.5% for the naive Bayes algorithm and MLP algorithm, respectively. However, in general, oversampling, which is a more costly sampling method, did not show a significant increase compared to the BRUS algorithm. Therefore, in the next stages of our study, we continued with the BRUS algorithm.

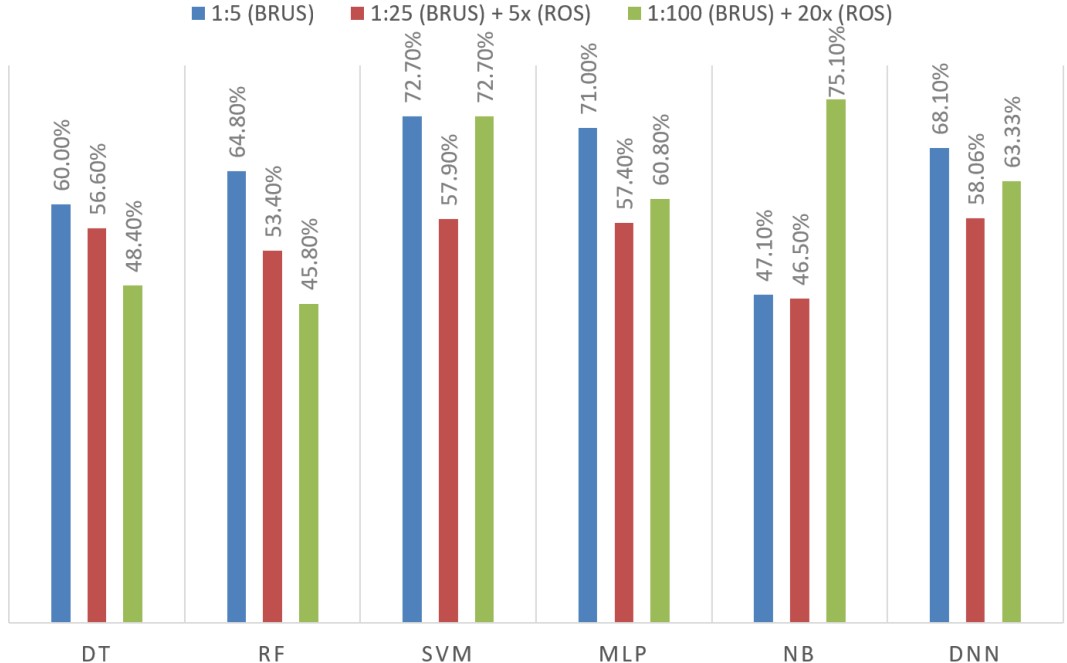

**Figure 4.** Balanced random undersampling (BRUS) vs. BRUS + random oversampling (ROS) in terms of F1 score.

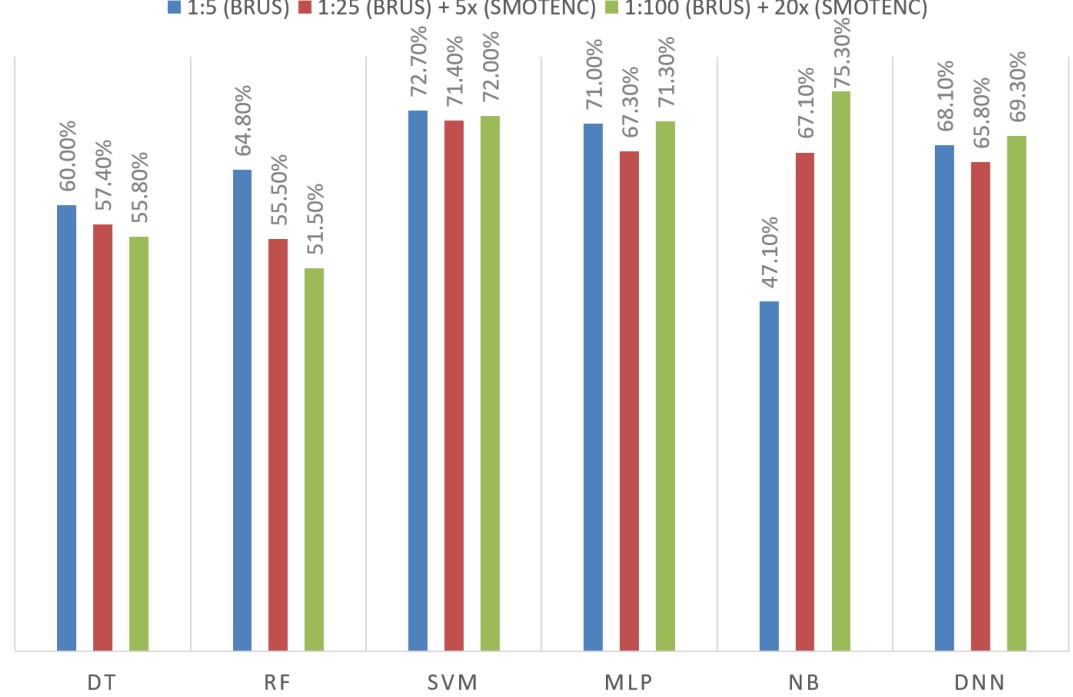

**Figure 5.** Balanced random undersampling (BRUS) vs. BRUS + SMOTENC in terms of F1 score.

To summarize, SVM outperforms other algorithms in terms of quantity-based evaluation, whereas DNN achieved great success against financial loss. On the other hand, we observed that the limit for transaction-based fraud detection in aviation stands at 80% in terms of F1 score for both quantity-based and cost-based measurement.

### 7.2. Profile-based Fraud Detection Results

To circumvent the limitations imposed by the transactional dataset and uncover historical patterns of passenger activity related to past transactions, a profile-based fraud-detection approach was introduced. This approach indirectly enhances the overall success of fraud detection. In this subsection, we demonstrate the test results of the proposed profiles for aviation and their contribution to the available transactional point of view.

Figures 6 and 7 show us that multi-modal profiling has come to the fore as the most successful profiling mode in general. In our study, the highest F1 score success in the transactional fraud-detection system was measured with the SVM algorithm at 77.8% on a quantity basis and with the DNN at 81.9% on a cost basis. However, the naive Bayes algorithm, utilizing multi-modal profiling, achieved a success of 88.2% and 92.2% on quantity basis and cost basis, respectively. A 7- to 17-point increase in the F1 score for the available algorithms encouraged us to introduce this profiling mechanism to the literature. Each profile contributed to this escalation, but both credit card profiling and record profiling made a big difference. On the other hand, we also observed that leaving transactional information out is not an option for a successful fraud-detection system, and utilizing only profiled information falls behind the multi-modal profiling mechanism. After we determined the multi-modal profiling as the most successful one in terms of both quantity and cost basis, comprehensive tests were performed using the multi-modal profiling mode.

Since new additional features are gathered from introduced profiles, we reran the feature-selection tests by also including Autoencoder due to its harmony with DNN. Contrary to our expectation, Figure 8 demonstrates that the CHI$^2$ algorithm is still leading and a 2.4% higher success rate was obtained compared to AutoEncoder and a 7.7% higher than PCA. Thus, CHI$^2$ was chosen for the feature-selection algorithm to be applied in the DNN tuning tests to be carried out in the next stage. We then conducted experiments for the new feature set obtained from both transactions and profiles, as given in Figure 9. The most successful algorithm then becomes naive Bayes, with an F1 score of 88.2%, whereas random forest, DNN, and MLP achieved a success rate of 86.6%, 85.3%, and 84.4%, respectively. The promising results of DNN encouraged us to examine the hyperparameters and improve the initial results. For this purpose, an experimental set was created and all tests in this experimental set were performed. Details of this experimental set and comprehensive results including accuracy, precision, and recall are given in Appendix A. The highest F1 score values were obtained in Experiment 45 on the basis of both quantity and cost. Quantity-based success and cost-based success reached 89.3% and 93.2%, respectively, and DNN takes the first place among others with the help of multi-modal profiling and hyperparameter adjustment. Analyzing the results thoroughly showed us that the network architecture and learning rate made the maximum contribution. It is also important to note that fine-tuning the hyperparameters is the essential step for such systems.

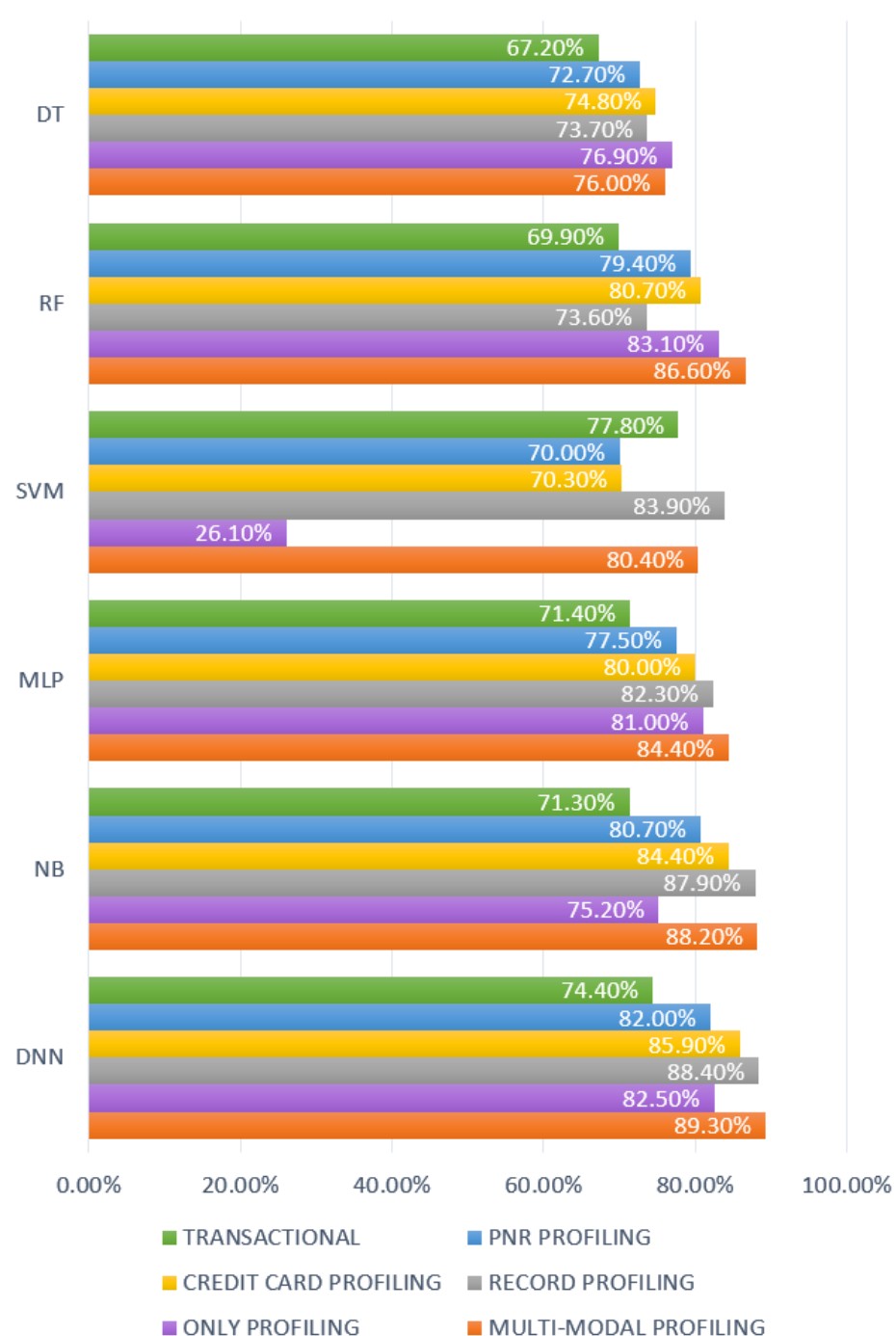

**Figure 6.** The contribution of the proposed profiling modes for each state-of-the-art algorithm on a quantity basis.

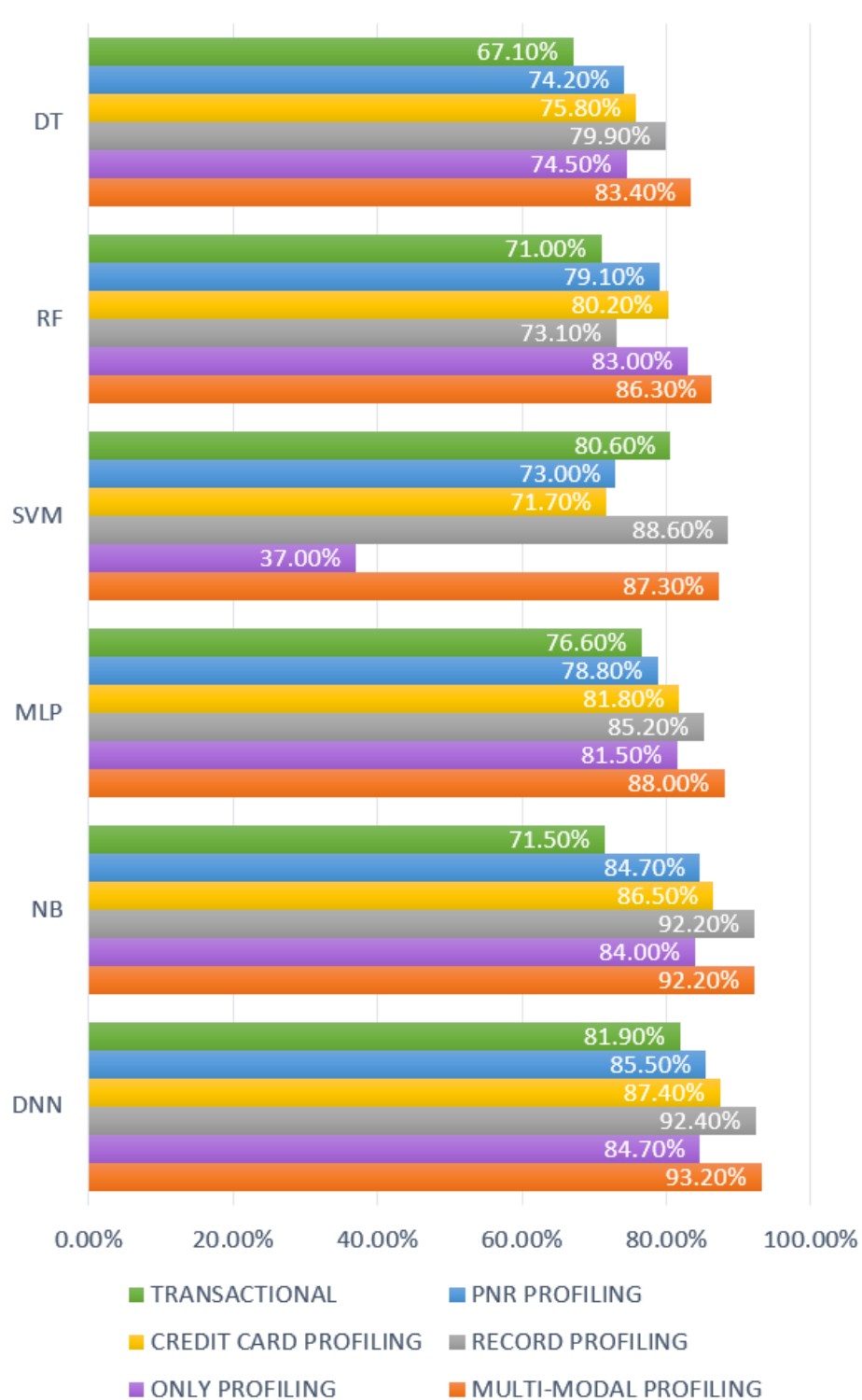

**Figure 7.** The contribution of the proposed profiling modes for each state-of-the-art algorithm on a cost basis.

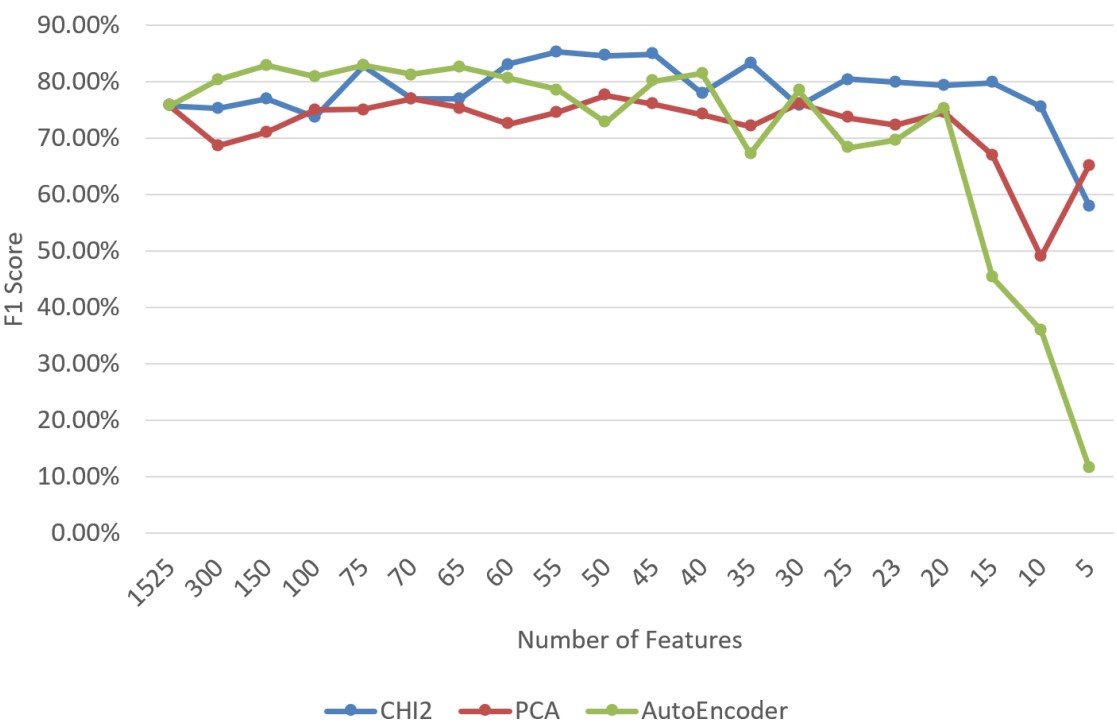

**Figure 8.** The impact of feature-selection algorithms on the performance of initial DNN utilizing multi-modal profiling.

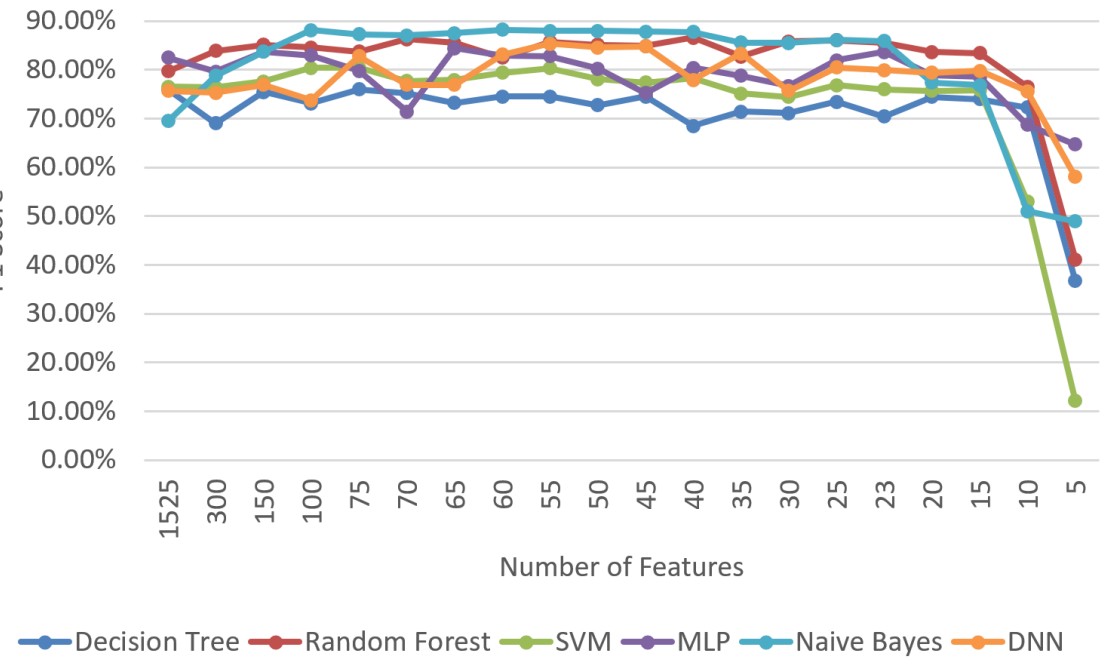

**Figure 9.** The impact of CHI$^2$ on the state-of-the-art algorithms utilizing multi-modal profiling.

In the tuning tests, we tried to determine the effect of the calibration of the hyperparameters on the system success. For this purpose, only one hyperparameter was changed each time, keeping all other test parameters constant, and the results were examined. As a result of this examination, we observed whether hyperparameters and architectural features have a direct impact on the result. In Figure 10, the number of epochs was changed between 50 and 500, and the relevant test results were examined. However, it was observed

that only the change in the epochs did not have a direct effect on the success. According to the learning rate value, more successful results can be obtained at different epoch numbers.

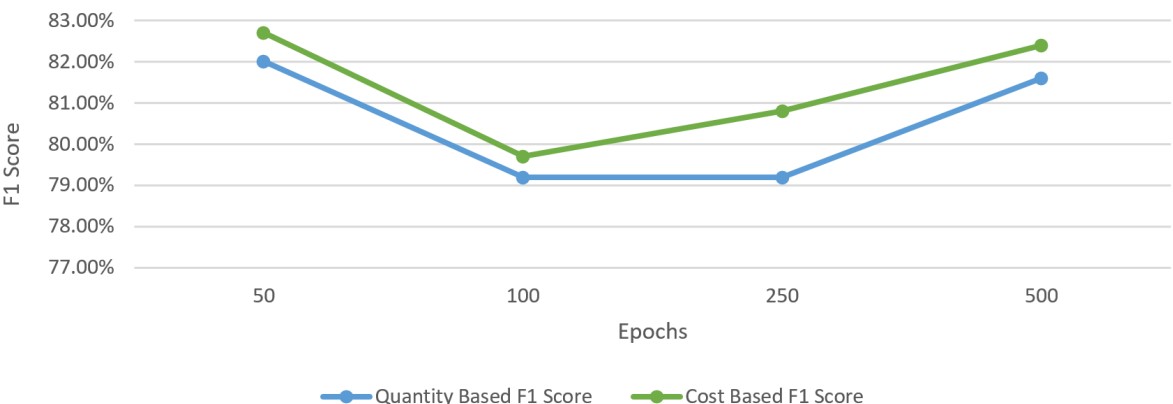

**Figure 10.** The impact of calibration of the number of epochs to the system success.

In batch size calibration tests, the results were examined by increasing the batch size by 2× between 32 and 512 each time. As can be seen in Figure 11, although there are some fluctuations, in general, the F1 score decreased by 2.9% on a quantity basis and 4% on a cost basis as a result of increasing the batch size. This calibration test set shows that higher success can be achieved with a lower batch size. However, it is important to achieve an optimum balance between batch size and training time in order to keep the training cost at an acceptable level.

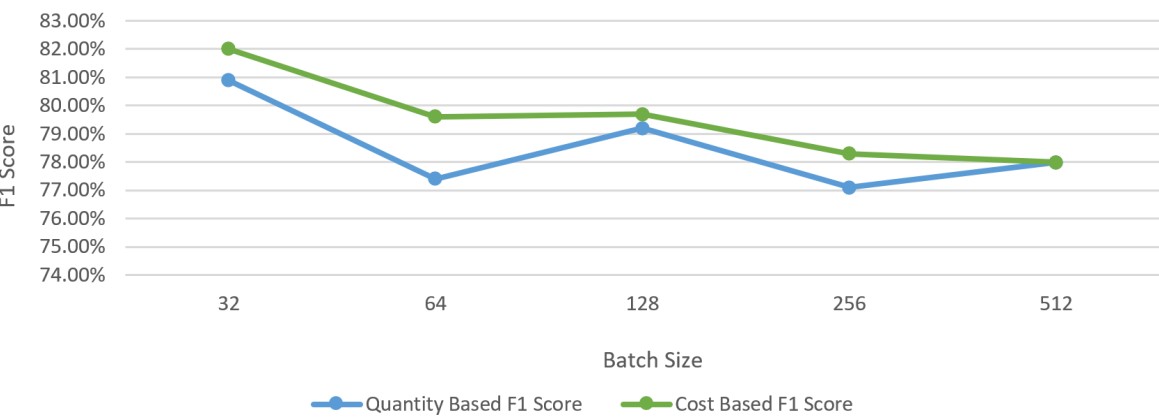

**Figure 11.** The impact of calibration of the batch size to the system success.

Although the effect of the learning rate hyperparameter on the result depends on parameters such as the batch size and epoch number, some calibration tests were carried out by keeping other variables constant. The results of these calibration tests are presented in Figure 12. As the learning rate decreased from 0.01 to 0.00001, the F1 score increased by 5.4% on a quantity basis and 8.8% on a cost basis. These results show that complex and detailed patterns are of great importance in the field of fraud detection for aviation.

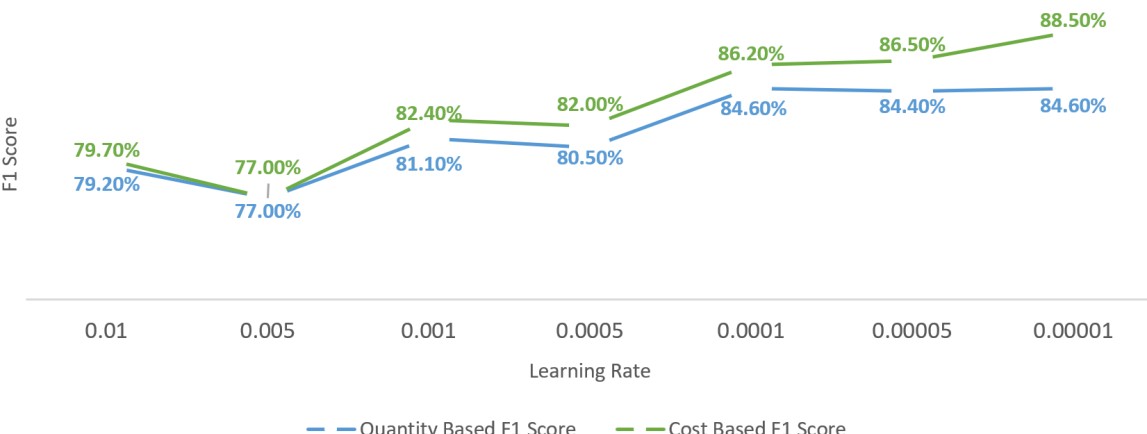

**Figure 12.** The impact of calibration of the learning rate to the system success.

In the DNN created in our study, we aimed to observe the success of dropout on hidden layers. Therefore, the best results between the tests with and without dropout are compared in Figure 13. As a result of this comparison, it was seen that the application of dropout on the DNN in our fraud-detection system has a positive effect on success. There was an F1 score increase of 3.2% on the basis of quantity and 3.5% on the basis of cost.

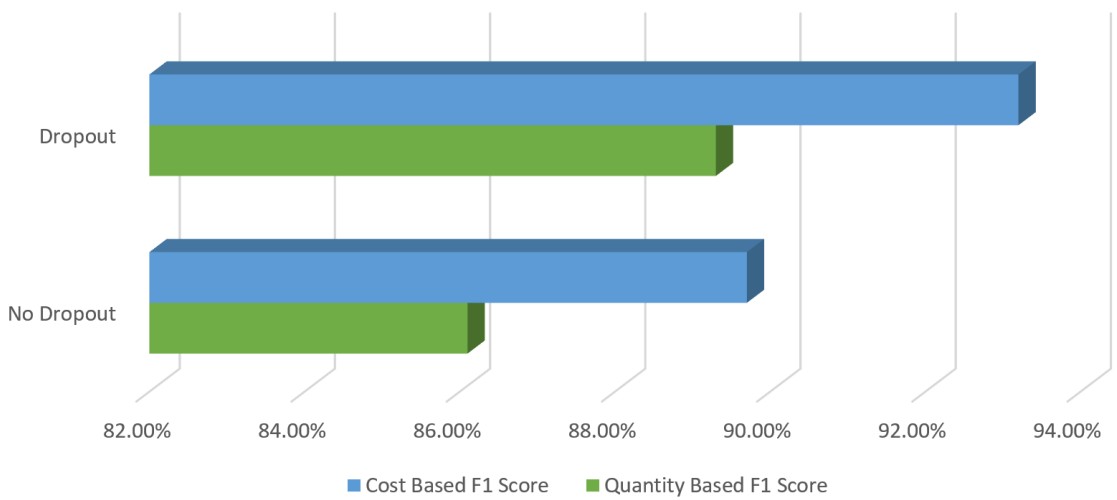

**Figure 13.** The impact of dropout on the DNN hidden layers on the system success.

Different DNN-layer formations are used in the calibration tests to observe the impact of DNN formation on the detection success. The results of these tests are given in increasing order according to the quantity-based F1 scores in Figure 14. The results clearly indicate that higher success rates are achieved in deep neural network (DNN) structures with more hidden layers. Upon examining the neuron configurations in these hidden layers, it becomes evident that success rates are higher when there is an increase in or a consistent number of neurons during layer transitions compared to other neuron formations.

As a result of the hyperparameter tuning tests, it was observed that the success rate could increase by 14.6% on the basis of the F1 score. Figure 15, where the test results are ordered from lowest to highest, shows us that a tuning stage performed on DNN has a critical importance for the fraud-detection system developed.

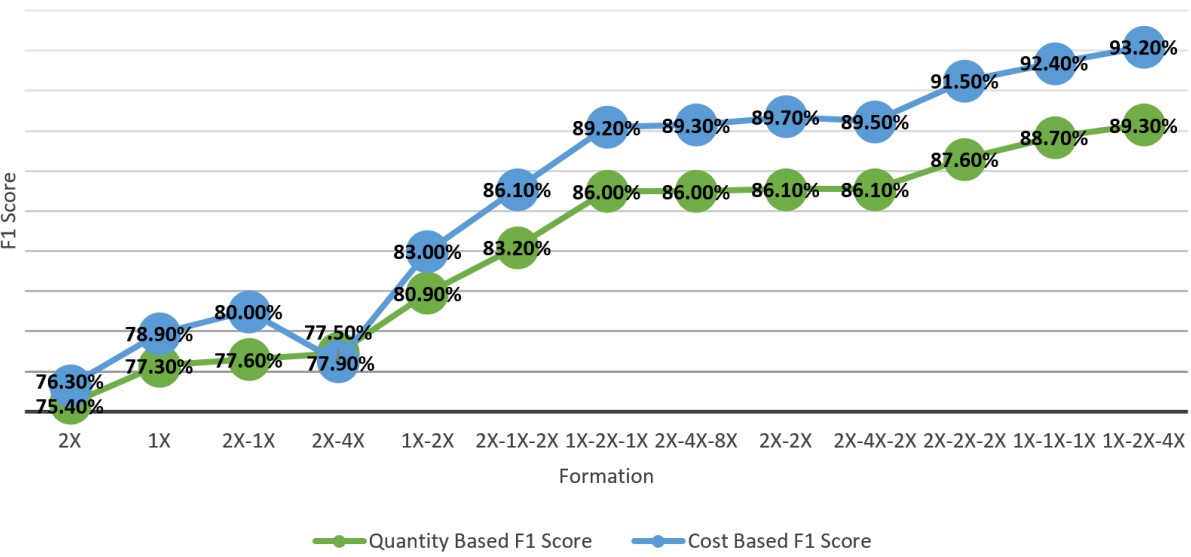

**Figure 14.** The impact of DNN layer formation to the system success.

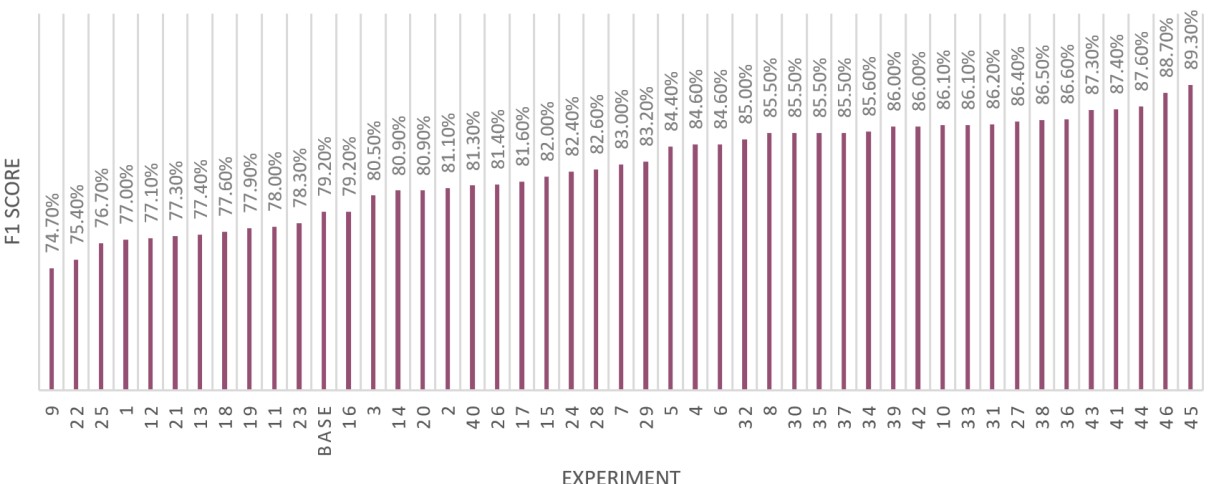

**Figure 15.** Ordered F1 score results of hyperparameter tuning experiments.

All the test results proved that the profile-based fraud-detection system performs significantly higher than the transaction-based fraud-detection system, as given in Table 6. The profile-based detection systems achieved approx. 90% and higher F1 scores for both quantity-based and cost-based measurements. The highest gain emerged via deep neural networks after the adjustment of hyperparameters. After deploying DNN with new hyperparameters, 18% and 21.7% improvements were observed on a quantity basis and on a cost basis, respectively.

**Table 6.** Transactional approach vs. profiled approach.

| Quantity-Based | | | | | |
|---|---|---|---|---|---|
| **Approach** | **Best Algorithm** | **Accuracy** | **Precision** | **Recall** | **F1 Score** |
| Transactional Approach—Initial | DNN | 91.00% | 86.90% | 54.60% | 67.10% |
| Transactional Approach—Improved | SVM | 93.60% | 79.00% | 76.20% | 77.60% |
| Profiled Approach—Initial | NB | 93.30% | 90.60% | 85.90% | 88.20% |
| Profiled Approach—Improved | DNN | 94.20% | 95.70% | 83.70% | 89.30% |
| Cost-Based | | | | | |
| **Approach** | **Best Algorithm** | **Accuracy** | **Precision** | **Recall** | **F1 Score** |
| Transactional Approach—Initial | DNN | 83.40% | 90.30% | 58.20% | 70.80% |
| Transactional Approach—Improved | DNN | 88.10% | 86% | 78.10% | 81.90% |
| Profiled Approach—Initial | NB | 91.90% | 90.70% | 93.80% | 92.20% |
| Profiled Approach—Improved | DNN | 92.20% | 96.20% | 90.40% | 93.20% |

## 8. Conclusions

Airline companies have considerably suffered from fraudulent activities in the last decade. Although many studies have aimed at detecting fraud operations, especially credit card fraud in the banking sector, to the best of our knowledge, this study is the initial and comprehensive one for fraudulent activities in aviation. We first gave a baseline for the success rate of the available state-of-the-art learning techniques. Then, several pre-processing techniques were applied to transactional airline sales data to improve the success, and the limit of transactional information for detecting suspicious airline ticket activities was determined. The necessity to benefit from the historical background of a passenger redirects us to creating profiles based on credit card, PNR, and record activities. Following this idea and combining this profiling mechanism with a deep learning approach enabled the proposed fraud system to achieve up to a 33.1% performance improvement in terms of F1 score, whereas we are able to decrease the financial loss by up to 31.6%. Although many fields in the transactional dataset contribute to the success, one of the indispensable fields for the cost-sensitive measurement and multi-modal profiling mechanisms developed in our study is the amount field. Thanks to the amount field, the cost-based success can be calculated and many statistical profile fields can be created. Our analysis showed that engineered features such as the number of chargebacks, e-mail addresses, phone numbers, and domestic and international flights are the prominent ones obtained from the credit card and PNR profiles. Additionally, we observed that information about minutes to flight time and the match status of passenger surname and cardholder surname revealed from record profiles are the salient features for detecting fraud transactions in aviation. We also remarked that fine-tuning the architecture of DNN regarding the aviation data could make a great difference of 14.6% in terms of the F1 score compared to randomly chosen hyperparameters or those taken from available studies about the banking sector. In the future, we are planning to utilize loss functions compatible with imbalanced data and/or to design a new one considering the cost-based gain. On the other hand, the proposed multi-modal profiling mechanism can also be applied to other sectors such as e-commerce and insurance where the data diversity is high. Only in sectors whose data structure is not suitable for multiple profiles, such as banking, a single-modal profiling mechanism based on credit card activities will need to be established instead of multi-modal.

**Author Contributions:** Conceptualization, M.T.A. and M.A.G.; methodology, M.T.A. and M.A.G.; software, M.T.A.; validation, M.T.A. and M.A.G.; formal analysis, M.T.A. and M.A.G.; investigation, M.T.A. and M.A.G.; resources, M.T.A.; data curation, M.T.A.; writing—original draft preparation, M.T.A.; writing—review and editing, M.A.G.; visualization, M.T.A.; supervision, M.A.G. All authors have read and agreed to the published version of the manuscript.

**Funding:** This work has received no external funding.

**Institutional Review Board Statement:** Not applicable.

**Informed Consent Statement:** Not applicable.

**Data Availability Statement:** The data presented in this study are available on request from the corresponding author. The data are not publicly available due to data is supplied by an airline company in Türkiye and do not give permission for making it publicly available.

**Acknowledgments:** We would like to sincerely thank Turkish Airlines company for sharing their data with us.

**Conflicts of Interest:** The authors declare no conflict of interest.

## Appendix A

A tuning experiment was carried out to examine the hyperparameters and improve the initial results. For this purpose, an experimental set was created, as given in Table A1. All tests in this experimental set were performed and the results are presented in Table A2.

**Table A1.** Experimental setup for tuning hyperparameters and network structure of DNN.

| Experiment | # of Hidden Layers | Layer Information | Learning Rate | Epoch | Batch Size | Dropout Rate | Input Dropout |
|---|---|---|---|---|---|---|---|
| Base Exp | 2 | 2x -2x | 0.01 | 100 | 128 | 0 | N/A |
| Exp-1 | 2 | 2x-2x | 0.005 | 100 | 128 | 0 | N/A |
| Exp-2 | 2 | 2x-2x | 0.001 | 100 | 128 | 0 | N/A |
| Exp-3 | 2 | 2x-2x | 0.0005 | 100 | 128 | 0 | N/A |
| Exp-4 | 2 | 2x-2x | 0.0001 | 100 | 128 | 0 | N/A |
| Exp-5 | 2 | 2x-2x | 0.00005 | 100 | 128 | 0 | N/A |
| Exp-6 | 2 | 2x-2x | 0.00001 | 100 | 128 | 0 | N/A |
| Exp-7 | 2 | 2x-2x | 0.00005 | 250 | 128 | 0 | N/A |
| Exp-8 | 2 | 2x-2x | 0.00001 | 250 | 128 | 0 | N/A |
| Exp-9 | 2 | 2x-2x | 0.00001 | 500 | 128 | 0 | N/A |
| Exp-10 | 2 | 2x-2x | 0.00001 | 500 | 128 | 0 | N/A |
| Exp-11 | 2 | 2x-2x | 0.01 | 100 | 512 | 0 | N/A |
| Exp-12 | 2 | 2x-2x | 0.01 | 100 | 256 | 0 | N/A |
| Exp-13 | 2 | 2x-2x | 0.01 | 100 | 64 | 0 | N/A |
| Exp-14 | 2 | 2x-2x | 0.01 | 100 | 32 | 0 | N/A |
| Exp-15 | 2 | 2x-2x | 0.01 | 50 | 128 | 0 | N/A |
| Exp-16 | 2 | 2x-2x | 0.01 | 250 | 128 | 0 | N/A |
| Exp-17 | 2 | 2x-2x | 0.01 | 500 | 128 | 0 | N/A |
| Exp-18 | 2 | 2x-x | 0.01 | 100 | 128 | 0 | N/A |
| Exp-19 | 2 | 2x-4x | 0.01 | 100 | 128 | 0 | N/A |
| Exp-20 | 2 | x-2x | 0.01 | 100 | 128 | 0 | N/A |
| Exp-21 | 2 | x | 0.01 | 100 | 128 | 0 | N/A |
| Exp-22 | 2 | 2x | 0.01 | 100 | 128 | 0 | N/A |
| Exp-23 | 3 | 2x-2x-2x | 0.01 | 100 | 128 | 0 | N/A |
| Exp-24 | 3 | 2x-4x-8x | 0.01 | 100 | 128 | 0 | N/A |
| Exp-25 | 3 | x-x-x | 0.01 | 100 | 128 | 0 | N/A |

**Table A1.** *Cont.*

| Experiment | # of Hidden Layers | Layer Information | Learning Rate | Epoch | Batch Size | Dropout Rate | Input Dropout |
|---|---|---|---|---|---|---|---|
| Exp-26 | 3 | x-2x-4x | 0.01 | 100 | 128 | 0 | N/A |
| Exp-27 | 3 | 2x-2x-2x | 0.01 | 100 | 128 | 0.2 | TRUE |
| Exp-28 | 3 | 2x-4x-8x | 0.01 | 100 | 128 | 0.2 | TRUE |
| Exp-29 | 3 | 2x-x-2x | 0.01 | 100 | 128 | 0.2 | TRUE |
| Exp-30 | 2 | 2x-2x | 0.00001 | 500 | 128 | 0.2 | TRUE |
| Exp-31 | 3 | 2x-2x-2x | 0.00001 | 500 | 128 | 0.2 | TRUE |
| Exp-32 | 3 | 2x-4x-8x | 0.00001 | 500 | 128 | 0.2 | TRUE |
| Exp-33 | 3 | 2x-4x-2x | 0.00001 | 500 | 128 | 0.2 | TRUE |
| Exp-34 | 3 | x-2x-4x | 0.00001 | 500 | 128 | 0.2 | TRUE |
| Exp-35 | 3 | x-2x-x | 0.00001 | 500 | 128 | 0.2 | TRUE |
| Exp-36 | 3 | x-x-x | 0.00001 | 500 | 128 | 0.2 | TRUE |
| Exp-37 | 2 | 2x-2x | 0.00001 | 500 | 128 | 0.2 | FALSE |
| Exp-38 | 3 | 2x-2x-2x | 0.00001 | 500 | 128 | 0.2 | FALSE |
| Exp-39 | 3 | 2x-4x-8x | 0.00001 | 500 | 128 | 0.2 | FALSE |
| Exp-40 | 3 | 2x-4x-2x | 0.00001 | 500 | 128 | 0.2 | FALSE |
| Exp-41 | 3 | x-2x-4x | 0.00001 | 500 | 128 | 0.2 | FALSE |
| Exp-42 | 3 | x-2x-x | 0.00001 | 500 | 128 | 0.2 | FALSE |
| Exp-43 | 3 | x-x-x | 0.00001 | 500 | 128 | 0.2 | FALSE |
| Exp-44 | 3 | 2x-2x-2x | 0.00002 | 500 | 128 | 0.3 | FALSE |
| **Exp-45** | **3** | **x-2x-4x** | **0.00002** | **500** | **128** | **0.3** | **FALSE** |
| Exp-46 | 3 | x-x-x | 0.00002 | 500 | 128 | 0.3 | FALSE |

**Table A2.** The experimental results of fine-tuning test setup for DNN.

| Experiment | Accuracy | C-Accuracy | Precision | C-Precision | Recall | C-Recall | F1 Score | C-F1-Score |
|---|---|---|---|---|---|---|---|---|
| Base Exp | 0.895 | 0.82 | 0.934 | 0.943 | 0.687 | 0.69 | 0.792 | 0.797 |
| Exp-1 | 0.886 | 0.8 | 0.931 | 0.937 | 0.656 | 0.654 | 0.77 | 0.77 |
| Exp-2 | 0.904 | 0.839 | 0.94 | 0.938 | 0.713 | 0.734 | 0.811 | 0.824 |
| Exp-3 | 0.901 | 0.836 | 0.937 | 0.934 | 0.707 | 0.73 | 0.805 | 0.82 |
| Exp-4 | 0.918 | 0.868 | 0.93 | 0.931 | 0.775 | 0.802 | 0.846 | 0.862 |
| Exp-5 | 0.918 | 0.872 | 0.947 | 0.942 | 0.761 | 0.799 | 0.844 | 0.865 |
| Exp-6 | 0.919 | 0.89 | 0.947 | 0.946 | 0.764 | 0.832 | 0.846 | 0.885 |
| Exp-7 | 0.911 | 0.854 | 0.93 | 0.931 | 0.75 | 0.772 | 0.83 | 0.844 |
| Exp-8 | 0.923 | 0.894 | 0.942 | 0.941 | 0.783 | 0.846 | 0.855 | 0.891 |
| Exp-9 | 0.879 | 0.785 | 0.955 | 0.953 | 0.614 | 0.61 | 0.747 | 0.744 |
| Exp-10 | 0.925 | 0.899 | 0.937 | 0.938 | 0.796 | 0.86 | 0.861 | 0.897 |
| Exp-11 | 0.891 | 0.808 | 0.938 | 0.942 | 0.668 | 0.665 | 0.78 | 0.78 |
| Exp-12 | 0.887 | 0.809 | 0.932 | 0.932 | 0.657 | 0.676 | 0.771 | 0.783 |
| Exp-13 | 0.888 | 0.819 | 0.936 | 0.939 | 0.66 | 0.691 | 0.774 | 0.796 |
| Exp-14 | 0.901 | 0.835 | 0.922 | 0.926 | 0.722 | 0.735 | 0.809 | 0.82 |
| Exp-15 | 0.905 | 0.84 | 0.919 | 0.93 | 0.74 | 0.745 | 0.82 | 0.827 |
| Exp-16 | 0.896 | 0.829 | 0.95 | 0.95 | 0.679 | 0.703 | 0.792 | 0.808 |
| Exp-17 | 0.904 | 0.841 | 0.924 | 0.944 | 0.731 | 0.732 | 0.816 | 0.824 |
| Exp-18 | 0.889 | 0.821 | 0.939 | 0.94 | 0.661 | 0.696 | 0.776 | 0.8 |
| Exp-19 | 0.89 | 0.803 | 0.934 | 0.935 | 0.668 | 0.661 | 0.779 | 0.775 |
| Exp-20 | 0.902 | 0.843 | 0.926 | 0.933 | 0.718 | 0.747 | 0.809 | 0.83 |

**Table A2.** *Cont.*

| Experiment | Accuracy | C-Accuracy | Precision | C-Precision | Recall | C-Recall | F1 Score | C-F1-Score |
|---|---|---|---|---|---|---|---|---|
| Exp-21 | 0.888 | 0.814 | 0.936 | 0.937 | 0.658 | 0.682 | 0.773 | 0.789 |
| Exp-22 | 0.881 | 0.797 | 0.944 | 0.945 | 0.627 | 0.64 | 0.754 | 0.763 |
| Exp-23 | 0.891 | 0.827 | 0.933 | 0.943 | 0.675 | 0.706 | 0.783 | 0.807 |
| Exp-24 | 0.908 | 0.851 | 0.929 | 0.933 | 0.74 | 0.763 | 0.824 | 0.84 |
| Exp-25 | 0.885 | 0.798 | 0.929 | 0.938 | 0.654 | 0.649 | 0.767 | 0.767 |
| Exp-26 | 0.904 | 0.842 | 0.928 | 0.93 | 0.724 | 0.748 | 0.814 | 0.829 |
| Exp-27 | 0.927 | 0.904 | 0.947 | 0.94 | 0.794 | 0.867 | 0.864 | 0.902 |
| Exp-28 | 0.911 | 0.856 | 0.947 | 0.942 | 0.732 | 0.765 | 0.826 | 0.844 |
| Exp-29 | 0.913 | 0.869 | 0.947 | 0.941 | 0.742 | 0.794 | 0.832 | 0.861 |
| Exp-30 | 0.923 | 0.897 | 0.948 | 0.943 | 0.779 | 0.85 | 0.855 | 0.895 |
| Exp-31 | 0.926 | 0.9 | 0.946 | 0.944 | 0.792 | 0.855 | 0.862 | 0.898 |
| Exp-32 | 0.921 | 0.892 | 0.946 | 0.944 | 0.772 | 0.839 | 0.85 | 0.888 |
| Exp-33 | 0.926 | 0.898 | 0.951 | 0.945 | 0.786 | 0.851 | 0.861 | 0.895 |
| Exp-34 | 0.924 | 0.896 | 0.946 | 0.944 | 0.782 | 0.847 | 0.856 | 0.893 |
| Exp-35 | 0.923 | 0.894 | 0.947 | 0.946 | 0.78 | 0.842 | 0.855 | 0.891 |
| Exp-36 | 0.928 | 0.9 | 0.943 | 0.938 | 0.8 | 0.862 | 0.866 | 0.899 |
| Exp-37 | 0.923 | 0.892 | 0.938 | 0.937 | 0.786 | 0.845 | 0.855 | 0.889 |
| Exp-38 | 0.927 | 0.905 | 0.938 | 0.939 | 0.803 | 0.87 | 0.865 | 0.903 |
| Exp-39 | 0.935 | 0.895 | 0.936 | 0.935 | 0.795 | 0.853 | 0.86 | 0.893 |
| Exp-40 | 0.904 | 0.848 | 0.934 | 0.938 | 0.72 | 0.754 | 0.813 | 0.836 |
| Exp-41 | 0.931 | 0.91 | 0.935 | 0.937 | 0.82 | 0.883 | 0.874 | 0.909 |
| Exp-42 | 0.925 | 0.894 | 0.935 | 0.933 | 0.796 | 0.854 | 0.86 | 0.892 |
| Exp-43 | 0.931 | 0.908 | 0.935 | 0.936 | 0.818 | 0.88 | 0.873 | 0.907 |
| Exp-44 | 0.933 | 0.922 | 0.951 | 0.952 | 0.812 | 0.881 | 0.876 | 0.915 |
| **Exp-45** | **0.942** | **0.922** | **0.957** | **0.962** | **0.837** | **0.904** | **0.893** | **0.932** |
| Exp-46 | 0.938 | 0.918 | 0.951 | 0.955 | 0.831 | 0.896 | 0.887 | 0.924 |

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
