# Peer review of "A Multi-Modal Profiling Fraud-Detection System for Capturing Suspicious Airline Ticket Activities"

_applsci, doi:10.3390/app132413121_

Round 1

Reviewer 1 Report

Comments and Suggestions for Authors

The article lacks the development of the literature review, the authors cite many studies, but there is no appropriate commentary on the cited items. What are the advantages and disadvantages of these methods, what is the research gap, what problems need to be solved - it is not clear.

There are block citations of several items, which means that it is not entirely clear what the authors are referring to. In my opinion, it is worth reviewing the literature more precisely.

The authors declare:

“In this study, novel solutions have been proposed for the essential problems encountered in fraud detection systems in the literature.”

But it is not fully explained what problems they solve, what were the proposals to solve these problems by other authors, what novelty was proposed in this study.

The article should strongly emphasize the scientific goal, not just the practical one, to define a scientific hypothesis. This should be in both the introduction and conclusion.

Empirical research, on the other hand, is presented in a correct and understandable way. Figures are clear and well complement the presented issues. The article is interesting and with the above changes it could be published.

Author Response

Reviewer#1, Concern # 1:

The article lacks the development of the literature review, the authors cite many studies, but there is no appropriate commentary on the cited items. What are the advantages and disadvantages of these methods, what is the research gap, what problems need to be solved - it is not clear.

Author response:  We thank the reviewer for raising this concern. We first went through our literature review section. We reread many of the cited studies and finally added appropriate commentary about the advantages and disadvantages of these studies.

Author action: We updated the Literature Review section by adding appropriate comments.

Reviewer#1, Concern # 2:

There are block citations of several items, which means that it is not entirely clear what the authors are referring to. In my opinion, it is worth reviewing the literature more precisely.

Author response:  We thank the reviewer for raising this concern. In our study, all processes and sub-mechanisms involved in a fraud detection system from start to end are discussed. In the literature review section, prominent studies on the basic issues in the fraud detection literature are explained and commented on in detail. However, in other sections where technical processes and mechanisms are discussed in detail, only studies using the same or similar approaches are cited as blocks. However, relevant sentences and paragraphs have been rearranged in the Literature Review section to clarify which technique or domain the block citations refer to.

Author action: Relevant sentences and paragraphs have been rearranged in the Literature Review section.

Reviewer#1, Concern # 3:

The authors declare:

“In this study, novel solutions have been proposed for the essential problems encountered in fraud detection systems in the literature.”

But it is not fully explained what problems they solve, what were the proposals to solve these problems by other authors, what novelty was proposed in this study.

Author response:  We thank the reviewer for pointing this out. We carefully rewrote the last part of the Literature Review section. We especially emphasize the novel parts of our study.

Author action:  We rewrote the last part of the Literature Review section.

“To the best of our knowledge, fraud detection problem in the aviation industry is hardly discussed and none of the comprehensive results were presented so far. In this study, we first give the baseline, then we select the appropriate instances using the proposed BRUS algorithm from the whole dataset to get rid of the negative effect of random selection. We additionally benefit from the oversampling technique and combine it with the BRUS algorithm to find the optimum ratio between fraud and non-fraud instances for a robust fraud detection model in aviation. We then introduce novice-engineered features from the proposed profiles using the historical connection of passenger activities. Finally, we evaluate the performance of our mechanism not just using classical metrics but also including cost-sensitive measurement.”

Reviewer#1, Concern # 4:

The article should strongly emphasize the scientific goal, not just the practical one, to define a scientific hypothesis. This should be in both the introduction and conclusion.

Author response:  We thank the reviewer for allowing us to clarify our study's scientific goals. Since there is no comprehensive study about fraud detection in aviation in the literature, we believe this study will be the first milestone in this domain. However, as the reviewer mentioned, the scientific goals of the study need to be clarified. Thus, we added the following paragraph to the Introduction section to define our two scientific goals.

Author action:  The following paragraph is added to the Introduction section.

 “The scientific goals of the fraud detection system developed in our study can be summarized in two essential points. First, we aim to maximize the F1 score, the most important performance metric for such imbalanced problems. Since a single and static profile structure has low success in fraud detection and many false positives, we build up historical connections using existing passenger activities obtained from the proposed profiles via feature engineering. The second goal of our study is to overcome the negative effect of the randomly selected instances as they might not represent the non-fraud data efficiently. Thus, we introduce the BRUS algorithm to include the most representative instances from the airline ticket dataset. This approach has a high negative sample representation ability as well as oversampling, has no misleading synthetic positive record, and provides ease of processing thanks to its smaller size.”

Reviewer#1, Concern # 5:

Empirical research, on the other hand, is presented in a correct and understandable way. Figures are clear and well complement the presented issues. The article is interesting and with the above changes it could be published.

Author response:  We thank the reviewer for his/her encouraging comments. We did our best to address all the concerns of the four reviewers.

Reviewer 2 Report

Comments and Suggestions for Authors

This paper introduce a novel multi-modal profiling mechanism based on deep learning for detection of fraudulent airline ticket activities in aviation considering the cost sensitive measurement. The main contribution is to construct new modals by profiling transaction data with chronological order and temporal flow features. The experimental results show that F1-score of the proposed system reaches up to 89.3% and 93.2% in terms of quantity-based success and cost-based success, respectively.

The only question is whether there is the loss of related transactions with the same customer or passenger in using BRUS sampling algorithm, which is not garantee the continuity of transaction data set.

Some character representation need to be modified such as line 504 ...are presented as in 1., and so on.

Whether the university of the proposed method may be further discussed and is applied to the other industries or domains.

Comments on the Quality of English Language

English language is good.

Reviewer 3 Report

Comments and Suggestions for Authors

This paper presents a deep network-based method for capturing suspicious airline ticket. In most cases, the paper is not written in the form of a scientific manuscript and it is not possible to follow the process of the article well.

The writing and language of the paper is not understandable.

Figures likes Fig.2 and Fig.3 are not prepared in the standard format.

The topic and innovation of the paper are not enough to be published in this journal.

Comments on the Quality of English Language

The writing and language of the paper is not understandable.

Reviewer 4 Report

Comments and Suggestions for Authors

1. The concepts of Accuracy & Precision may be better explained indicating the difference.

2. You may explain any salient feature of the data for which the techniques used work better.

3. Page 17: why the gain on NB after using the state of the art learning algorithms, is less? Any intuitive explanation?
